# Uni-PrevPredMap: Extending PrevPredMap to a Unified Framework of Prior-Informed Modeling for Online Vectorized HD Map Construction

## Abstract

Safety constitutes a foundational imperative for autonomous driving systems, necessitating maximal incorporation of accessible prior information. This study establishes that temporal perception buffers and cost-efficient high-definition (HD) maps inherently form complementary prior sources for online vectorized HD map construction. We present Uni-PrevPredMap, a pioneering unified framework systematically integrating previous predictions with corrupted HD maps. Our framework introduces a tri-mode paradigm maintaining operational consistency across non-prior, temporal-prior, and temporal-map-fusion modes. This tri-mode paradigm simultaneously decouples the framework from ideal map assumptions while ensuring robust performance in both map-present and map-absent scenarios. Additionally, we develop a tile-indexed 3D vectorized global map processor enabling efficient 3D prior data refreshment, compact storage, and real-time retrieval. Uni-PrevPredMap achieves state-of-the-art map-absent performance across established online vectorized HD map construction benchmarks. When provided with corrupted HD maps, it exhibits robust capabilities in error-resilient prior fusion, empirically confirming the synergistic complementarity between temporal predictions and imperfect map data. Code is available in supplementary materials.

## 1 Introduction

High-Definition (HD) maps serve as critical infrastructure for autonomous vehicles, delivering centimeter-level road geometry and semantic information to ensure precise localization and safe navigation. These maps can be generated through two primary approaches: traditional offline SLAM-based mapping workflows or emerging online perception systems. Given the prohibitive costs associated with offline HD map production and maintenance, the automotive industry is increasingly prioritizing online vectorized HD map construction techniques Liao et al. (2023); Jiang et al. (2023).

Online vectorized HD map construction experiences reliability degradation when onboard sensors encounter visual deprivation scenarios, including severe occlusion environments and beyond-visual-range conditions. To address these limitations, researchers have proposed utilizing two prior sources: temporal perception buffers and cost-efficient HD maps (e.g., less frequently updated HD maps and crowd-sourced HD maps). The two prior sources exhibit inherent complementarity: temporal buffers ensure backward continuity through sequential observation accumulation, while cost-efficient HD maps establish forward-looking constraints via predefined geometry structures. However, existing prior-informed models remain limited to processing either temporal perception buffers Yuan et al. (2024a); Wang et al. (2024b); Peng et al. (2024); Chen et al. (2024); Zhang et al. (2024a); Kim et al. (2025) or cost-efficient HD maps Jiang et al. (2024); Sun et al. (2023) in isolation, as depicted in Figure 1 (a). Consequently, designing a unified framework compatible with both priors is crucial to fully leverage their strengths and enhance performance for safety-critical autonomous systems.

Integrating temporal perception buffers with cost-efficient HD map priors presents two fundamental challenges: (1) Cost-efficient HD maps are prone to errors, including false positives, false negatives, and pose misalignment. Current HD map prior models shown in Figure 1 (b), however, tend to

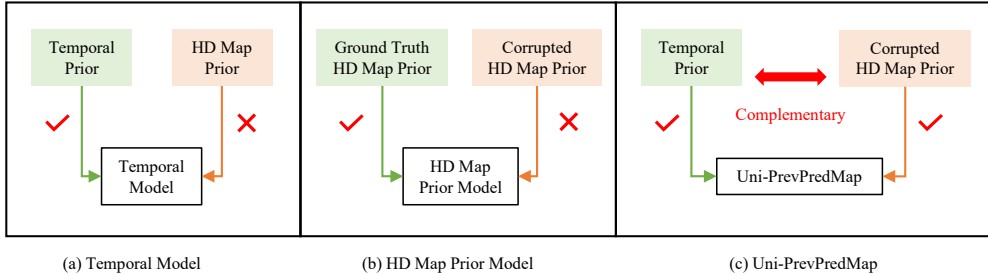

Figure 1: Motivation of Uni-PrevPredMap. (a) Existing prior-informed models are constrained to handling temporal perception buffers or cost-efficient HD maps separately. (b) Current HD map prior models tend to over-rely on idealized fidelity assumption. (c) Uni-PrevPredMap delivers robust standalone perception in map-absent areas while boosting reliability in map-present areas through adaptive fusion coupled with active map error identification.

over-rely on idealized fidelity assumptions Bateman et al. (2024), causing potential ground truth leakage. To mitigate this, the unified framework must actively detect map errors rather than implicitly trust flawed priors. (2) Cost-efficient HD maps inevitably contain uncovered regions. Consequently, the unified framework must exhibit robust standalone perception capabilities in map-absent areas, while simultaneously enhancing reliability in map-present areas through adaptive fusion.

In this work, we present Uni-PrevPredMap, extending the temporal baseline PrevPredMap Peng et al. (2024), as a unified prior-informed framework for online vectorized HD map construction. Our approach explores the previously untapped synergy between two complementary prior sources: previous predictions and corrupted HD maps. To address existing challenges, we introduce a tri-mode paradigm employing stratified sampling across non-prior, temporal-prior, and temporal-map-fusion modes during training, where map priors undergo instance-level perturbation. This simple but effective paradigm simultaneously decouples the model from ideal map assumptions while enabling robust performance in both map-absent and map-present scenarios. Furthermore, we incorporate a novel tile-indexed 3D vectorized global map processor enabling efficient 3D prior updates, compact storage, and real-time retrieval. This processor uses geolocation-specific tile partitioning synchronized with vehicle positioning to eliminate complex post-processing. The solution extends existing 2D rasterized Zhang et al. (2024a) and vectorized Shi et al. (2024) approaches into full 3D vectorization, enhancing real-world applicability.

In summary, the contributions of this work include: (1) We establish that temporal perception buffers and cost-efficient HD maps serve as complementary priors for online vectorized HD map construction. Based on this foundational insight, we propose Uni-PrevPredMap, the first unified framework to strategically integrate previous predictions with corrupted HD maps. (2) A tri-mode paradigm is introduced to preserve operational consistency across non-prior, temporal-prior, and temporal-map-fusion modes, simultaneously decoupling Uni-PrevPredMap from ideal map assumptions while maintaining robust performance in both map-absent and map-present scenarios. (3) A tile-indexed 3D vectorized global map processor is developed to enable efficient 3D prior data refreshment, compact storage, and real-time retrieval. (4) Uni-PrevPredMap achieves state-of-the-art map-absent performance across established online vectorized HD map construction benchmarks. When processing corrupted HD maps, it demonstrates robust error-resilient prior fusion capabilities, empirically confirming the synergistic complementarity between temporal predictions and cost-efficient HD maps.

## 2 RELATED WORK

Online vectorized HD map construction was initially conceptualized as a semantic segmentation problem Li et al. (2022a); Pan et al. (2020); Chen et al. (2022); Li et al. (2022b). HDMapNet Li et al. (2022a) established a raster-to-vector conversion pipeline that first generates BEV semantic segmentation maps and subsequently groups these pixel-wise results into vectorized instances through heuristic post-processing. VectorMapNet Liu et al. (2023) introduced the first end-to-end framework, utilizing an auto-regressive transformer architecture for sequential vector instance retrieval. MapTR

Liao et al. (2023) subsequently revolutionized this domain through a unified permutation-equivalent representation and hierarchical query embedding scheme, achieving one-stage parallel decoding that significantly enhanced computational efficiency. Recent advancements demonstrate following innovation directions, including concise map representations Ding et al. (2023); Li et al. (2024); Qiao et al. (2023); Zhou et al. (2024); Zhang et al. (2024b); Liu & Yuan (2024), optimized attention mechanisms Liao et al. (2024); Hu et al. (2024); Xu et al. (2024), structural query designs Liu et al. (2024b); Xu et al. (2024), multi-modal distillation Hao et al. (2024), and segmentation-based auxiliary supervision Liao et al. (2024); Liu et al. (2024a); Choi et al. (2024); Ma et al. (2024).

## 2.1 ONLINE VECTORIZED HD MAP CONSTRUCTION WITH TEMPORAL PERCEPTION BUFFERS

Runtime temporal perception information inherently exists without supplementary acquisition overhead. In temporal modeling methodologies, StreamMapNet Yuan et al. (2024a) implements dense-sparse feature co-fusion through streaming integration of BEV and query features. SQD-MapNet Wang et al. (2024b) advances this paradigm by introducing a stream query denoising strategy to facilitate temporal consistency learning. MapTracker Chen et al. (2024) and MapUnveiler Kim et al. (2025) leverage memory buffers to integrate previous BEV and query features, achieving temporal fusion at the feature level. In contrast, PrevPredMap Peng et al. (2024) pioneers a prediction-level temporal modeling approach, implying distinct advantages for seamless map integration. Building upon this foundation, Uni-PrevPredMap extends PrevPredMap Peng et al. (2024) into a unified framework that effectively incorporates two complementary prior sources: historical predictions and imperfect HD maps.

Generally, map elements exhibit static properties. Map features or predictions perceived at location X can serve as prior whenever the road structure near X remains unchanged. Building on this observation, NMP Xiong et al. (2023) and NeMO Zhu et al. (2023) develop region-centric approaches that leverage temporal information. PreSight Yuan et al. (2024b) introduces Neural Radiance Fields (NeRF) to alleviate memory constraints and generate city-scale priors. HRMapNet Zhang et al. (2024a) employs a global map processor that stores and distributes rasterized historical predictions. Uni-PrevPredMap redesigns this component into a tile-indexed 3D vectorized representation. This redesign achieves three key improvements: (1) elevating the representation from 2D to 3D, (2) reducing storage overhead by shifting from rasterization to vectorization, and (3) enhancing retrieval efficiency through tile-based indexing. This processor is central to our framework, maintaining global state for both temporal buffer and corrupted maps.

## 2.2 ONLINE VECTORIZED HD MAP CONSTRUCTION WITH COST-EFFICIENT ALTERNATIVE MAPS

Cost-efficient maps provide critical priors for online vectorized HD map construction. Multiple prior categories have been explored, including Standard Definition (SD) maps Luo et al. (2024); Jiang et al. (2024); Wu et al. (2024), HD maps Jiang et al. (2024); Sun et al. (2023), and satellite imagery Gao et al. (2024). Although SD maps are widely available, they lack granular semantics required for safety-critical autonomy. These maps cannot encode essential divider attributes like color and type, nor can they represent pedestrian crossings, stop lines, directional arrows, or elevation data. While cost-efficient HD maps present higher accessibility barriers than SD maps, their operational feasibility remains proven: major OEMs routinely maintain lightweight HD maps containing basic lane geometry and safety-critical elements. However, current HD map prior models often rely heavily on idealized map assumptions, leading to significant performance degradation when processing erroneous map data Bateman et al. (2024), a phenomenon that suggests potential ground truth leakage. In contrast to these methods, which assume that the prior map is geometrically accurate, semantically complete, and fully covering the operational area, Uni-PrevPredMap is designed to be robust to these deviations. It effectively handles geometric noise, missing or extraneous map elements, and uncovered regions, thereby reducing dependence on ideal map conditions.

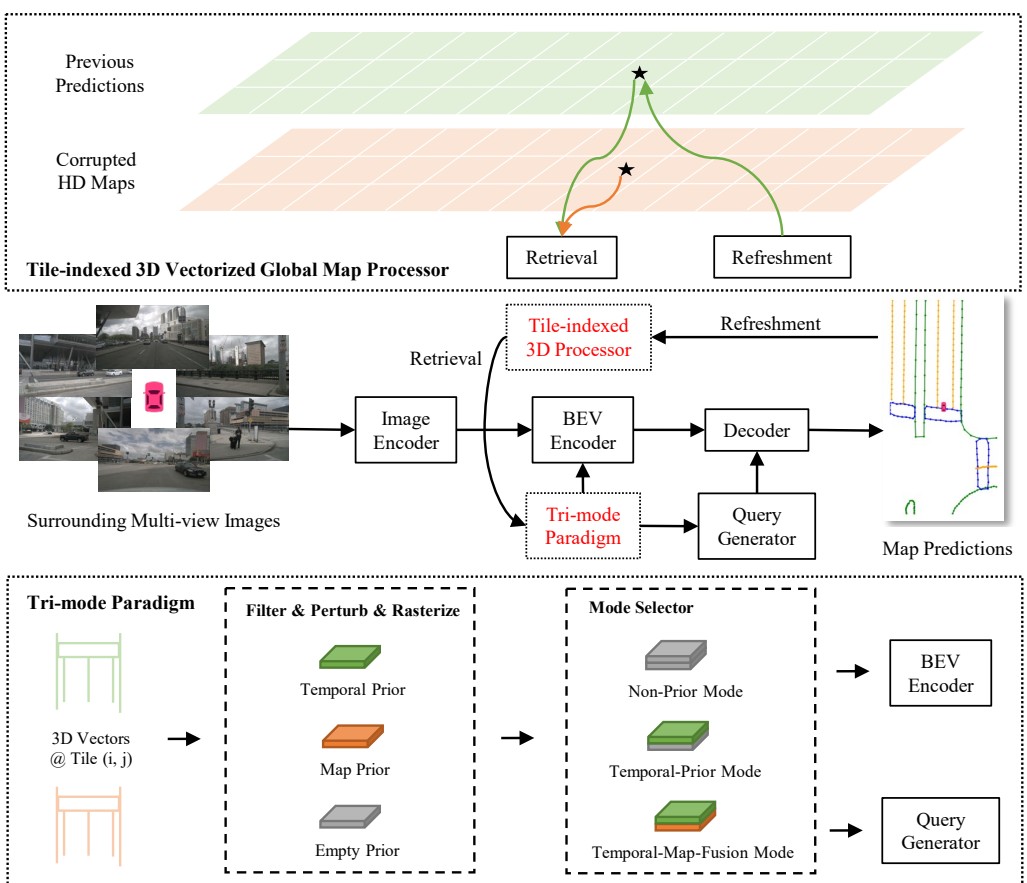

Figure 2: The overall architecture of Uni-PrevPredMap. The upper dashed region highlights the tile-indexed 3D vectorized global map processor enabling efficient 3D prior data refreshment, storage, and retrieval. The lower dashed region outlines the tri-mode paradigm ensuring consistent operation across non-prior, temporal-prior, and temporal-map-fusion modes.

## 3 METHOD

### 3.1 OVERALL ARCHITECTURE

Uni-PrevPredMap retains PrevPredMap's core architectural principle of prediction-driven temporal modeling instead of feature-level processing, enabling native map prior assimilation. Figure 2 illustrates the overall architecture. Surrounding multi-view images are processed through an image encoder to extract features. Concurrently, the tile-indexed 3D vectorized global map processor retrieves temporal and map priors using vehicle positioning coordinates. Through the tri-mode paradigm, these priors are processed into non-prior, temporal-prior, or temporal-map-fusion configurations based on the selected mode. The processed priors then combine with image features via the BEV encoder to generate prior-augmented BEV features while simultaneously initializing queries. These prior-informed BEV and query features are subsequently decoded into final map predictions that undergo incremental updates within the tile-indexed 3D vectorized global map processor.

### 3.2 TRI-MODE PARADIGM

We introduce a tri-mode paradigm ensuring operational robustness across non-prior, temporal-prior, and temporal-map-fusion modes. Through instance-level perturbation applied to map priors, this tri-mode paradigm simultaneously decouples the model from ideal map assumptions while enabling consistent performance in both map-absent and map-present scenarios.

As illustrated in Figure 2 and Algorithm 1 (appendix), retrieved 3D vectors first undergo spatial filtering constrained by intersection with the predefined perception range. Crucially, 2D vectors inherently lack vertical differentiation capability, preventing distinction of multi-level configurations like elevated overpasses and subterranean tunnels that risk propagating erroneous priors. During training, instance-level perturbation is applied to filtered map priors, where random vector subsets receive distinct displacement magnitudes sampled uniformly from [0,6] meters. This stochastically displaces approximately half of map vectors with mean 3-meter displacement, aligning with standard lane widths. The process simulates real-world infrastructure modifications including divider adjustments, boundary changes, and pedestrian crossing relocation. During inference, Uni-PrevPredMap's robustness is tested against untrained perturbations including instance-level addition/deletion and frame-level displacement/rotation/scaling operations, with detailed definitions and visualizations in Appendix A.1. Finally, rasterization generates temporal, map, and empty priors.

According to the selected operational mode, temporal, map, and empty priors are combined pairwise before concatenation as shown in Figure 2. During training, Uni-PrevPredMap employs a stratified sampling strategy across three distinct modes: non-prior, temporal-prior, and temporal-map-fusion. These modes are sampled according to a fixed ratio. In the temporal-map-fusion mode, map vectors are retrieved from both the temporal buffer and corrupted map of the global map. These vectors are rasterized into temporal and map heatmaps before being processed by the BEV encoder and query generator. The temporal-prior mode uses only vectors from the temporal buffer to generate the temporal heatmap, while the map heatmap is set to a zero matrix. In the non-prior mode, both heatmaps are initialized as zero matrices. This tri-mode design enables seamless switching during inference based on the availability of cost-effective HD maps. The system adaptively activates the temporal-map-fusion mode when operating continuously within a mapped area with both temporal buffer and map data available. The temporal-prior mode is triggered upon entering a region with no map coverage, relying solely on the temporal buffer. The non-prior mode serves as a fallback when both the temporal buffer is unavailable (e.g., after a system restart) and map data is absent.

Processed priors then follow two parallel processing pathways: (1) The BEV encoder concatenates priors with BEV features using HRMapNet's approach Zhang et al. (2024a), followed by convolutional layer fusion; (2) The query generator employs deformable attention where priors function as keys/values for spatial interaction with map queries.

### 3.3 Tile-indexed 3D Vectorized Global Map Processor

We propose a tile-indexed 3D vectorized global map processor that eliminates complex post-processing through geolocation-synchronized tile partitioning, enabling efficient 3D prior updates, compact storage, and real-time retrieval. Specifically, the tile-indexed processor implements dual-axis geospatial indexing where each (i,j)-indexed tile defines a bounded geographical region containing discrete map vectors confined to respective tile boundaries.

**Refreshment** selectively integrates high-confidence predictions into ego-pose-associated tiles of the global map, with detailed mapping in Appendix A.2. For previous predictions, predicted vectors are updated to specified tiles, whose indices are calculated using the vehicle's UTM (Universal Transverse Mercator) coordinates. For corrupted HD maps, global vectors are pre-stored in corresponding indexed tiles for target regions, with tile indices derived from the map vector's UTM coordinates.

**Retrieval** queries relevant map elements from specific tiles anchored to the ego pose. Temporal and map priors are retrieved from target tiles indexed via the vehicle's UTM coordinates, leveraging the refreshment mapping described above. To ensure data integrity, adjacent tiles surrounding the target tile are concurrently retrieved, with detailed visualizations and mapping in Appendix A.2.

Table 1: Comparison with SOTA methods on nuScenes. All backbones utilized are ResNet50. FPS measurements are conducted on the same machine with NVIDIA RTX A6000. ⋆ are taken from the corresponding papers and are scaled based on the FPS of MapTRv2 Liao et al. (2024).

| Method | Epoch | $AP_{div}$ | $AP_{ped}$ | $AP_{bou}$ | mAP | FPS |
|---|---|---|---|---|---|---|
| MapTRv2 Liao et al. (2024) | 24 | 62.4 | 59.8 | 62.4 | 61.3 | 16.4 |
| HRMapNet Zhang et al. (2024a) | 24 | 67.4 | 65.8 | 68.5 | 67.2 | 13.3 |
| Mask2Map Choi et al. (2024) | 24 | 71.3 | 70.6 | 72.9 | 71.6 | 9.5 |
| MapTracker Chen et al. (2024) | 24 | 69.2 | 75.3 | 71.2 | 71.9 | 11.7 |
| MapUnveiler Kim et al. (2025) | 24 | 67.6 | 67.6 | 68.8 | 68.0 | 13.4⋆ |
| PriorMapNet Wang et al. (2024a) | 24 | 69.0 | 64.0 | 68.2 | 67.1 | 13.7⋆ |
| FastMap Hu et al. (2025) | 24 | 69.1 | 65.5 | 69.7 | 68.1 | 17.2⋆ |
| Uni-PrevPredMap | 24 | **72.3** | **76.2** | **73.6** | **74.0** | 12.2 |
| Uni-PrevPredMap* | 24 | 84.1 | 79.1 | 79.6 | 80.9 | 11.5 |
| MapTRv2 Liao et al. (2024) | 110 | 68.8 | 68.0 | 71.0 | 69.2 | 16.4 |
| HIMap Zhou et al. (2024) | 110 | 75.0 | 71.3 | 74.7 | 73.7 | 10.0⋆ |
| HRMapNet Zhang et al. (2024a) | 110 | 72.9 | 72.0 | 75.8 | 73.6 | 13.3 |
| Mask2Map Choi et al. (2024) | 110 | 73.6 | 73.1 | **77.3** | 74.6 | 9.5 |
| MapTracker Chen et al. (2024) | 72 | 74.1 | **80.0** | 74.1 | 76.1 | 11.7 |
| MGMapNet Yang et al. (2024) | 110 | 74.3 | 71.8 | 74.8 | 73.6 | 13.3⋆ |
| Uni-PrevPredMap | 72 | **76.3** | 77.9 | 76.6 | **77.0** | 12.2 |
| Uni-PrevPredMap* | 72 | 83.4 | 79.7 | 82.3 | 81.8 | 11.5 |

# 4 EXPERIMENT

## 4.1 EXPERIMENTAL SETUP

**Datasets**  We evaluate Uni-PrevPredMap on two popular and large-scale datasets: nuScenes Caesar et al. (2020) and Argoverse2 Wilson et al. (2023). The nuScenes dataset offers 2D vectorized maps alongside 1000 scenes, with 700 designated for training and 150 for validation. Each scene encompasses 20 seconds of 2Hz RGB images captured by 6 cameras. Argoverse2, on the other hand, delivers 3D vectorized maps and consists of 1000 logs, with 700 allocated for training and 150 for validation. Each log comprises 15 seconds of 20Hz RGB images from 7 ring cameras.

**Evaluation Metrics**  Consistent with previous methods Li et al. (2022a); Liu et al. (2023); Liao et al. (2023), we select three static map categories for a fair evaluation: pedestrian crossings, lane dividers, and road boundaries. The perception range is set as 30m front and rear and 15m left and right of the vehicle. The common average precision (AP) based on Chamfer Distance is used as the evaluation metric under 3 threholds of {0.5, 1.0, 1.5}m.

**Implementation Details**  We utilize ResNet50 He et al. (2016) as the perspective backbone and LSS-based BEVPoolv2 Huang & Huang (2022) as the parameterized PV-to-BEV transformation network. The optimizer is AdamW with a weight decay 0.01, and the initial learning rate is set to 0.0006, employing a cosine decay schedule. The batch size is 16 and all models are trained with 4 NVIDIA A100 GPUs. We define the size of each BEV grid as 0.3 meters. The default numbers of instance queries, point queries and decoder layers are 100, 20 and 6, respectively.

## 4.2 COMPARISONS WITH STATE-OF-THE-ART METHODS

**Performance on nuScenes**  As shown in Table 1, Uni-PrevPredMap achieves 74.0 mAP (24-epoch training) and 77.0 mAP (72-epoch training), surpassing all SOTA methods in training convergence, validation accuracy, and inference speed. When integrated with corrupted HD maps, the enhanced variant Uni-PrevPredMap* attains 80.9 mAP (24-epoch) and 81.8 mAP (72-epoch), demonstrating stable performance under map degradation. While Uni-PrevPredMap demonstrates strong overall performance, it does not surpass every state-of-the-art method across all metrics, particularly in boundary and pedestrian crossing. Under the map-absent setting, Uni-PrevPredMap achieves a Boundary AP

Table 2: Comparison with SOTA methods on Argoverse2. All backbones utilized are ResNet50.

| Method | $AP_{div}$ | $AP_{ped}$ | $AP_{bou}$ | mAP |
|---|---|---|---|---|
| MapTRv2 Liao et al. (2024) | 68.9 | 60.7 | 64.5 | 64.7 |
| HIMap Zhou et al. (2024) | 68.3 | 66.7 | 70.3 | 68.4 |
| MapUnveiler Kim et al. (2025) | 72.6 | 66.0 | 67.6 | 68.7 |
| MGMapNet Yang et al. (2024) | 72.1 | 64.7 | 70.4 | 69.1 |
| PriorMapNet Wang et al. (2024a) | 73.4 | 66.5 | 69.8 | 69.9 |
| Uni-PrevPredMap | **74.5** | **69.5** | **72.9** | **72.3** |
| Uni-PrevPredMap* | 82.5 | 75.2 | 80.5 | 79.4 |

Table 3: Framework evolution: PrevPredMap to Uni-PrevPredMap.

| Setting | mAP (w/o map) | mAP (w/ map) |
|---|---|---|
| PrevPredMap | 66.3 | - |
| + tile-indexed 3D vectorized global map processor | 74.0 | - |
| + tri-mode paradigm | 74.0 | 80.9 |
| = Uni-PrevPredMap | +7.7 | +80.9 |

that is 0.7 lower than Mask2Map Choi et al. (2024). We attribute this primarily to the difference in conditioning strategies: Mask2Map Choi et al. (2024) utilizes the segmentation heatmap of the current frame as prior, which provides highly precise and up-to-date spatial information. This proves particularly advantageous for modeling boundaries, which tend to exhibit significant inter-frame variation. In contrast, our approach relies on historical heatmaps, leading to a slight decrease in Boundary AP in the absence of a map. When a corrupted map prior is provided, Uni-PrevPredMap surpasses Mask2Map Choi et al. (2024) by a significant margin of 5.0 AP. In the case of pedestrian crossings, Uni-PrevPredMap underperforms MapTracker Chen et al. (2024) by 2.1 AP under the map-absent setting. This can be explained by the fact that pedestrian crossings are less structured and more challenging to reconstruct from rasterized heatmaps compared to dividers and boundaries. MapTracker Chen et al. (2024) leverages historical query features that preserve vector-level semantics, thereby reducing the modeling difficulty for such elements. However, when a noisy map prior is introduced, the performance gap narrows to just 0.3 AP. Inspired by these two observations, we identify two promising directions for future work: incorporating the segmentation heatmap of the current frame to enhance spatial precision, and integrating vector-level semantic features to better model unstructured elements. Any such enhancements, however, must carefully balance the potential performance gains against the increase in inference latency caused by added model complexity.

**Performance on Argoverse 2** Argoverse2 provides a 3D vectorized map containing additional elevation data, addressing the vertical dimension information lacking in the nuScenes dataset. As demonstrated in Table 2, comparative evaluations under 3D evaluation configurations reveal performance characteristics. Following the experimental protocol established by MapTRv2, we trained Uni-PrevPredMap over 6 epochs and evaluated it at a 2.5Hz sampling rate. The results in Table 2 indicate that Uni-PrevPredMap achieves 72.3 mAP, outperforming all SOTA methods in 3D vectorized HD map construction on the Argoverse2 benchmark. While Uni-PrevPredMap*, enhanced through integration with corrupted HD maps, achieves superior performance with 79.4 mAP.

### 4.3 ABLATION STUDY

**Framework Evolution: PrevPredMap to Uni-PrevPredMap** Based on PrevPredMap Peng et al. (2024), we incrementally introduce a tile-indexed 3D vectorized global map processor and tri-mode paradigm, with results detailed in Table 3. The 3D processor enables multi-frame historical predictions to inform the model through both BEV encoder and query generator pathways, extending beyond PrevPredMap's single-frame query attention mechanism. This advancement increases map-absent

Table 4: Comparative performance across tri-mode paradigm variants.

| Training Paradigm | mAP (w/o map) | mAP (w/ map) |
|---|---|---|
| Single-mode | 16.6 | 82.9 |
| Dual-mode-a | 60.6 | 77.4 |
| Dual-mode-b | 69.0 | 84.4 |
| Tri-mode | 74.0 | 80.9 |

Table 5: Inference performance under different prior conditions.

| Temporal Prior | Map Prior | mAP | FPS |
|---|---|---|---|
| ✗ | ✗ | 64.9 | 14.2 |
| ✓ | ✗ | 74.0 | 12.2 |
| ✗ | ✓ | 71.3 | 13.1 |
| ✓ | ✓ | 80.9 | 11.5 |

Table 6: Robustness to artificial HD map perturbations (Inst. = Instance-level, Frame = Frame-level).

| Type | Noise Range | mAP |
|---|---|---|
| Perfect Map | - | 85.6 |
| Inst. Displacement | [-3m,3m] | 81.9 |
| | [-6m,6m] | 80.9 |
| | [-9m,9m] | 80.3 |
| Inst. Addition | [0,10] | 85.2 |
| Inst. Deletion | [0,10] | 82.5 |
| Frame Displacement | [-3m,3m] | 77.0 |
| | [-6m,6m] | 75.7 |
| | [-9m,9m] | 74.6 |
| Frame Rotation | [-15°,15°] | 77.0 |
| Frame Scale | [0.8,1.2] | 77.0 |
| Empty Map | - | 74.0 |

performance from 66.3 to 74.0 mAP. Crucially, the tri-mode paradigm unlocks robust map usage, elevating performance to 80.9 mAP with a map prior while maintaining the 74.0 mAP competitive performance without it. This demonstrates that our model gains a significant advantage when a map is available, without any degradation when it is absent.

**Comparative Performance across Tri-mode Paradigm Variants** Table 4 systematically explores degraded variants of the proposed tri-mode paradigm. The single-mode variant trains exclusively under map-prior conditions, while dual-mode configurations include dual-mode-a (non-prior and map-prior modes) and dual-mode-b (temporal-prior and temporal-map-fusion modes). As shown, the single-mode approach, which is typical in existing HD map prior models Jiang et al. (2024); Sun et al. (2023), collapses in map-absent scenarios. We emphasize that robust map-absent perception capability serves as a necessary condition to prevent ground truth leakage. Performance gaps reveal critical dependencies: the 13.4 mAP deficit in temporal-deprived dual-mode-a demonstrates synergistic necessity between temporal buffers and cost-efficient HD maps, whereas the 5.0 mAP shortfall in dual-mode-b underscores integrated adaptability's indispensability. Essentially, the tri-mode paradigm proves pivotal by liberating models from idealized map assumptions while achieving state-of-the-art map-absent performance and significantly boosting accuracy with cost-efficient HD maps. These advances address core challenges in online vectorized HD map construction: maintaining robust standalone operation in map-absent regions and enhancing reliability through active map error identification in map-present areas.

**Inference Performance under Different Prior Conditions** Table 5 demonstrates that our tri-mode paradigm ensures consistent robustness across map-absent (74.0 mAP) and map-present (80.9 mAP) scenarios. Combined temporal and map priors (80.9 mAP) significantly outperform individual priors (temporal: 74.0 mAP; map: 71.3 mAP), confirming their complementary role in enhancing perception stability through temporal buffering and HD map constraints.

**Robustness to Artificial HD Map Perturbations** Uni-PrevPredMap is trained with instance-level displacement uniformly sampled from [0, 6m], while more unseen perturbations are applied during inference to simulate real-world corruption and test robustness. Specifically, we implement instance-level addition/deletion and frame-level displacement/rotation/scaling operations, with detailed definitions and visualizations in Appendix A.1. As Table 6 shows, even with perfect maps, Uni-PrevPredMap achieves 85.6 mAP, significantly below perfect accuracy, indicating it leverages map priors through map-absent perception capability (74.0 mAP) rather than memorizing ground truth. Performance adaptively adjusts to displacement ranges: smaller deviations show improvement while larger ones cause deterioration. Instance-level addition and deletion respectively simulate road removals and unmapped constructions, with Uni-PrevPredMap demonstrating exceptional resilience

Table 7: Ablation on the tri-mode sampling ratio with T:T&M fixed at 3:2 on nuScenes. "N", "T", and "M" denote non-prior, temporal prior, and map prior respectively.

| Ratio (N:T:T&M) | mAP (T) | mAP (M) | mAP (T&M) |
|---|---|---|---|
| 0.30 : 0.42 : 0.28 | 73.4 | 72.4 | 82.7 |
| 0.40 : 0.36 : 0.24 | 73.7 | 71.8 | 82.0 |
| 0.50 : 0.30 : 0.20 | 74.0 | 71.3 | 80.9 |
| 0.60 : 0.24 : 0.16 | 73.9 | 70.3 | 79.7 |
| 0.70 : 0.18 : 0.12 | 72.7 | 69.4 | 78.0 |

Table 8: Ablation on the tri-mode sampling ratio with N:(T+T&M) fixed at 1:1 on nuScenes.

| Ratio (N:T:T&M) | mAP (T) | mAP (M) | mAP (T&M) |
|---|---|---|---|
| 0.50 : 0.35 : 0.15 | 73.1 | 69.8 | 79.4 |
| 0.50 : 0.30 : 0.20 | 74.0 | 71.3 | 80.9 |
| 0.50 : 0.25 : 0.25 | 73.6 | 72.0 | 81.4 |
| 0.50 : 0.50 : 0.00 | 74.0 | - | - |

Table 9: Ablation on the tri-mode sampling ratio with N:(T+T&M) fixed at 1:1 on ArgoVerse2.

| Ratio (N:T:T&M) | mAP (T) | mAP (M) | mAP (T&M) |
|---|---|---|---|
| 0.50 : 0.35 : 0.15 | 71.5 | 72.9 | 77.8 |
| 0.50 : 0.30 : 0.20 | 72.4 | 73.6 | 78.6 |
| 0.50 : 0.25 : 0.25 | 72.3 | 74.7 | 79.4 |
| 0.50 : 0.50 : 0.00 | 71.9 | - | - |

(merely 0.4 mAP drop for additions, controlled 3.1 mAP drop for deletions). We emphasize that HD map prior models must discriminatively leverage both existence and non-existence evidence to enhance reliable cues while suppressing erroneous ones, thereby optimizing scene understanding. For frame-level perturbations where all map vectors uniformly deviate from temporal observations, Uni-PrevPredMap maintains robustness against corrupted HD maps, outperforming map-absent baselines. This resilience likely stems from cross-validation mechanisms between temporal observations and map constraints that effectively mitigate erroneous prior impacts.

**Ablation on the Sampling Ratio of the Tri-mode Paradigm** During training, Uni-PrevPredMap employs stratified sampling across non-prior, temporal-prior, and temporal-map-fusion modes. We performed ablations from two perspectives. First, with the ratio between temporal-prior (T) and temporal-map fusion (T&M) modes fixed at 3:2, we varied the proportion of the non-prior mode (N) to the combined prior-based modes (T+T&M). As shown in Table 7, the best map-absent performance was achieved at N:(T+T&M) = 0.5:0.5. Reducing the proportion of (T+T&M) led the model to overfit the non-prior scenario, degrading performance in both temporal-prior and fusion-prior settings. Conversely, increasing (T+T&M) resulted in over-reliance on priors: while fusion-prior performance improved slightly, temporal-prior performance declined. This may be attributed to the fact that temporal-prior inference still relies on an initial non-prior prediction; weakening that foundation adversely affects subsequent predictions. Second, with N:(T+T&M) fixed at 1:1, we varied the T:T&M ratio on both nuScenes (Table 8) and Argoverse2 (Table 9). The results showed consistent trends across datasets, supporting the generalizability of the design. The optimal map-absent performance was observed at N:T:T&M = 0.5:0.3:0.2. Increasing T&M led to over-reliance on map priors, slightly degrading temporal-prior performance. Reducing T&M, while avoiding such over-reliance, could still harm temporal modeling if map priors provide useful supervisory signals. Thus, insufficient map involvement also impaired performance. When T&M was entirely absent, the model reverted to a temporal-only mode.

**Qualitative Analysis** Figure 3 shows predictions and priors from Uni-PrevPredMap, comparing results with and without corrupted HD maps in occlusion, rainy, and night conditions. Purple shading highlights the differences. Without corrupted maps, the model fails to accurately predict map elements in these challenging environments, whereas the incorporation of corrupted HD maps enables effective use of complementary priors, resulting in significantly clearer and more precise predictions. Additional examples can be found in Appendix A.7.

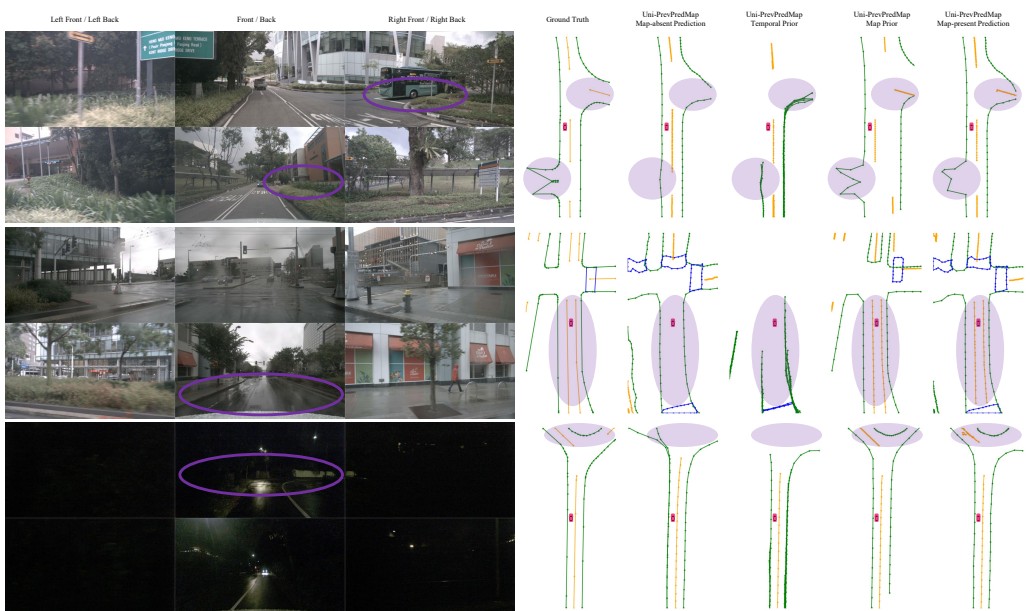

Figure 3: Comparison of predictions by Uni-PrevPredMap with versus without corrupted HD maps, with differences highlighted in purple shading across occlusion, rainy, and night scenarios. Corresponding priors are illustrated to demonstrate their influence. Green, orange, and blue lines represent road boundaries, lane dividers, and pedestrian crossings, respectively.

### 4.4 LIMITATIONS AND FUTURE WORK

Based on current understanding, Uni-PrevPredMap's limitations and future work focus on three key aspects. (1) The 3D global map uses height data only for filtering, not prior generation, due to the high latency of 3D rasterization, and limited vertical diversity in existing benchmarks. Future work should focus on efficient 3D representations and richer 3D datasets. (2) Evaluation relied on simulated perturbations; a main limitation is the absence of real corrupted maps (e.g., from outdated or crowdsourced sources). Building standardized real-corruption benchmarks is an essential next step. (3) Integrating temporal priors with complementary information shows promise for end-to-end autonomous driving. Map construction inherently functions as a fundamental auxiliary task, while 3D object detection could leverage vehicle-infrastructure cooperation to acquire prior information about surrounding entities. Incorporating such complementary priors may enhance robustness and safety in complex scenarios.

## 5 CONCLUSION

This paper introduces Uni-PrevPredMap, a unified prior-informed framework for online vectorized HD map construction that integrates two complementary prior sources: previous predictions and corrupted HD maps. The framework employs a tri-mode paradigm ensuring operational consistency across non-prior, temporal-prior, and temporal-map-fusion modes. This tri-mode paradigm simultaneously liberates the system from ideal map assumptions while maintaining robust performance in both map-present and map-absent scenarios. Complementing this, we develop a tile-indexed 3D vectorized global map processor enabling efficient 3D prior updates, compact storage, and real-time retrieval. Uni-PrevPredMap achieves state-of-the-art performance on established online vectorized HD map benchmarks without map priors. When processing corrupted HD maps, it demonstrates robust error-resilient fusion capabilities that empirically validate the synergistic complementarity between temporal predictions and imperfect map data. We anticipate this work will provide valuable insights for autonomous driving perception systems.

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

# A   APPENDIX

---

**Algorithm 1:** Tri-mode Paradigm during Training

---

**Input:** Mode sampling ratio $R$, global map $G(i, j)$, ego pose $E$
**Output:** Temporal heatmap $H_t$, map heatmap $H_m$

1   $H_t, H_m \leftarrow$ zeros();
2   $mode \leftarrow$ sampleMode($R$);
3   **if** $mode == $ *non-prior* **then**
4     |   **return** $H_t, H_m$
5   **end**
6   **else**
7     |   $V_t \leftarrow$ retrieveVectors($G(i, j), E$);
8     |   $H_t \leftarrow$ rasterizeToHeatmap($V_t$);
9     |   **if** $mode == $ *temporal-map-fusion* **then**
10     |    |   $V_m \leftarrow$ retrieveVectors($G(i, j), E$);
11     |    |   $H_m \leftarrow$ rasterizeToHeatmap($V_m$);
12     |   **end**
13     |   **return** $H_t, H_m$
14   **end**

---

## A.1   ARTIFICIAL HD MAP PERTURBATIONS

We design six perturbation types to simulate real-world corruption and test robustness, including instance-level displacement, addition, and deletion alongside frame-level displacement, rotation, and scaling. Comparative visualizations of these perturbations are provided in Figure 4.

**Instance-level displacement** applies distinct displacement magnitudes to randomly selected map vector subsets per frame, sampling magnitudes uniformly over a specified range. For Uni-PrevPredMap training, this range is specifically set to [-6, 6] meters, stochastically displacing statistically averaged 50% of map vectors with mean 3-meter displacement approximating standard lane width specifications. This configuration simulates common infrastructure modifications including divider addition/removal, boundary adjustments, and pedestrian crossing relocation through controlled perturbation protocols.

**Instance-level addition** incorporates $n \sim U[0, 10]$ non-existent elements per frame to simulate real-world map element removals such as road closures or infrastructure decommissioning. These elements are randomly sampled from all map elements across the entire validation dataset, ensuring structurally diverse yet geometrically plausible configurations.

**Instance-level deletion** removes $n \sim U[0, min(10, n_{gt})]$ exising elements per frame to simulate unmapped construction scenarios, where $n_{gt}$ denotes the map element count in the corresponding frame. Given that nuScenes datasets average fewer than 10 map elements per frame, this configuration statistically deletes an average of 50% of map vectors per frame.

**Frame-level displacement** uniformly displaces all map vectors per frame by sampling displacement magnitudes from a uniform distribution over a specified range, specifically simulating positional errors within pose misalignment scenarios.

**Frame-level rotation** uniformly rotates all map vectors per frame through sampling $\theta \sim U[-15°, 15°]$, specifically simulating orientation errors within pose misalignment scenarios.

**Frame-level scaling** uniformly scales all map vectors per frame through sampling $s \sim U[0.8, 1.2]$, inherently generating displacement as a geometric coupling effect due to coordinate transformation.

## A.2   REFRESHMENT AND RETRIEVAL OF TILE-INDEXED 3D VECTORIZED GLOBAL MAP PROCESSOR

Algorithms 2 and 3 illustrate the workflows of the refreshment and retrieval modules, respectively.

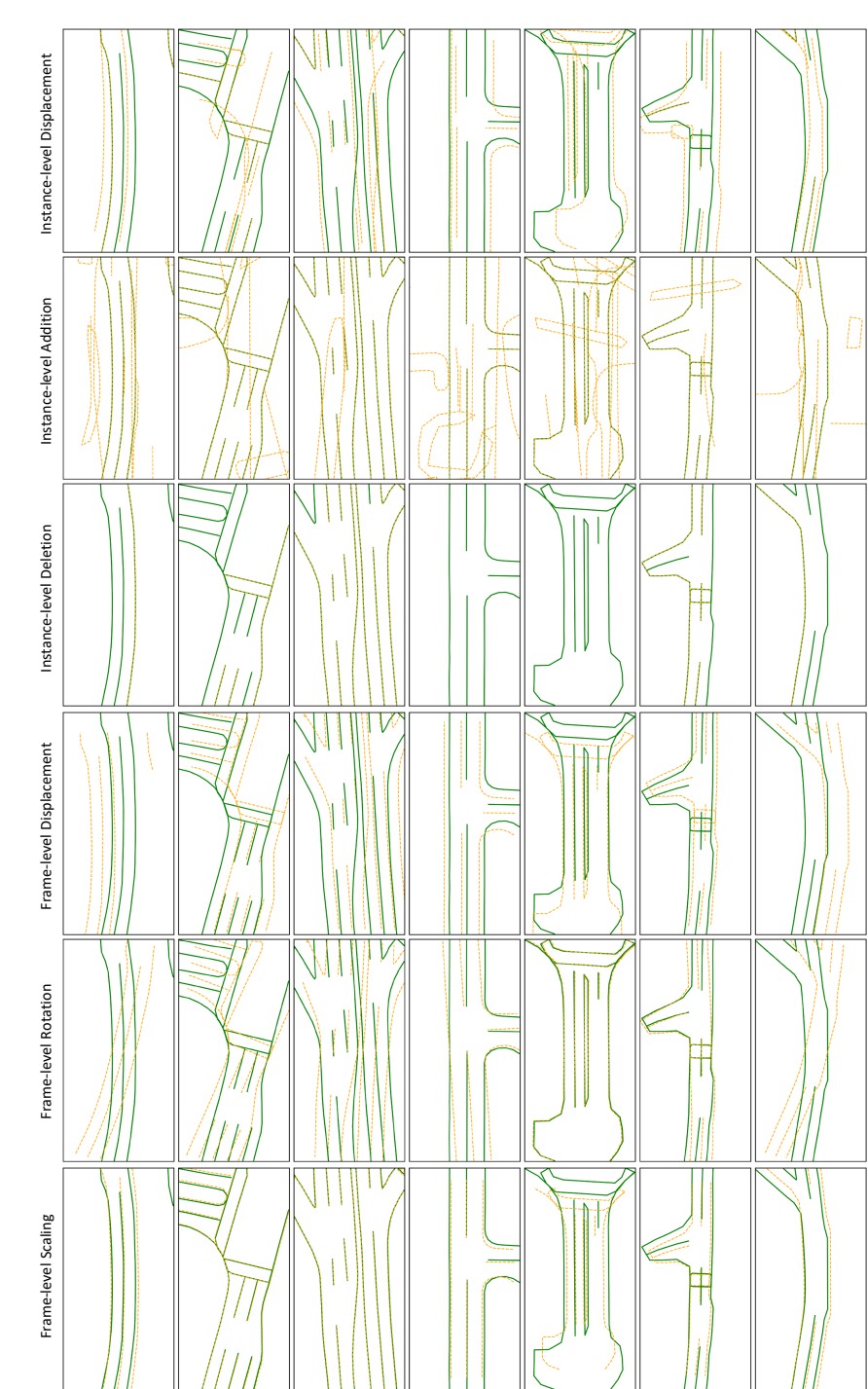

Figure 4: Comparison of artificially perturbed HD maps (orange dashed) vs. ground truth annotations (green solid) with identical random seeds ensuring reproducible and fair comparisons.

**Refreshment** mapping between tile indices $(i, j)$ and UTM coordinates $(UTM_{east}, UTM_{north})$ is defined as:

$$(i, j) = (UTM_{east}, UTM_{north})//l, \tag{1}$$

where $l$ denotes the long side length of the perception range.

|  |  |  |
|---|---|---|
| $(i_t\text{-}1, j_t\text{+}1)$ | $(i_t, j_t\text{+}1)$ | $(i_t\text{+}1, j_t\text{+}1)$ |
| $(i_t\text{-}1, j_t)$ | ⭐ | $(i_t\text{+}1, j_t)$ |
| $(i_t\text{-}1, j_t\text{-}1)$ | $(i_t, j_t\text{-}1)$ | $(i_t\text{+}1, j_t\text{-}1)$ |

Figure 5: Visualization of adjacency selection during retrieval. The central grid indicates the target tile with indices $(i_t, j_t)$. Orange and green star markers denote distinct vehicle UTM coordinate positions within the target tile, with corresponding shaded grids indicating respective adjacent tiles.

**Retrieval** mapping between tile indices $\{(i, j) \mid i \in I, \ j \in J\}$ and UTM coordinates $(UTM_{east}, UTM_{north})$ is defined as:

$$
I = \begin{cases} (i_t - 1, i_t) & if \quad UTM_{east}\%l < l/2 \\ (i_t) & if \quad UTM_{east}\%l = l/2 \\ (i_t, i_t + 1) & if \quad UTM_{east}\%l > l/2 \end{cases} , \tag{2}
$$

$$
J = \begin{cases} (j_t - 1, j_t) & if \quad UTM_{north}\%l < l/2 \\ (j_t) & if \quad UTM_{north}\%l = l/2 \\ (j_t, j_t + 1) & if \quad UTM_{north}\%l > l/2 \end{cases} , \tag{3}
$$

where $i_t$ and $j_t$ denote the indices of target tile. As depicted in Figure 5, the adjacency selection is determined by the spatial position of the vehicle's UTM coordinates relative to the target tile boundaries.

---

**Algorithm 2:** Refreshment Algorithm

---

**Input:** Global map $G(i, j)$, predicted map vectors $V(k)$, confidence threshold $\tau$, ego pose $E$
**Output:** Updated global map $G(i, j)$
1   $i, j \leftarrow \text{getTileIndex}(E)$;
2   **for** *each map vector k in $V(k)$* **do**
3     **if** *confidence$(V(k)) > \tau$* **then**
4       $\tilde{V}(k) \leftarrow \text{egoToGlobal}(V(k), E)$;
5       $G(i, j) \leftarrow G(i, j) \parallel \tilde{V}(k)$;
6     **end**
7   **end**

---

**Algorithm 3:** Retrieval Algorithm

---

**Input:** Global map $G(i, j)$, ego pose $E$, perception range $R$
**Output:** Retrieved map vectors $V(k)$
1   $i, j \leftarrow \text{getTileIndex}(E)$;
2   $list_i, list_j \leftarrow \text{getAdjacentTileIndexList}(E, R)$;
3   $list_i, list_j \leftarrow list_i \parallel i, list_j \parallel j$;
4   **for** *each i in $list_i$* **do**
5     **for** *each j in $list_j$* **do**
6       $\tilde{V}(\hat{k}) \leftarrow \text{filterByPerceptionRange}(G(i, j), R)$;
7       $V(\hat{k}) \leftarrow \text{globalToEgo}(\tilde{V}(\hat{k}), E)$;
8       $V(k) \leftarrow V(k) \parallel V(\hat{k})$;
9     **end**
10   **end**

### A.3 COMPARISON ON GEOGRAPHICALLY NON-OVERLAPPING NUSCENES AND ARGOVERSE2

StreamMapNet Yuan et al. (2024a) proposes a geographically non-overlapping dataset partitioning strategy for nuScenes and ArgoVerse2. The comparative performance of Uni-PrevPredMap under this configuration is detailed in Table 10. As evidenced by Table 10, Uni-PrevPredMap maintains superior performance in both temporal-prior and temporal-map-fusion-prior operational modes under the geographically non-overlapping data distribution paradigm.

Table 10: Comparison on geographically non-overlapping nuScenes and ArgoVerse2.

| Dataset | Method | $AP_{div}$ | $AP_{ped}$ | $AP_{bou}$ | mAP |
|---|---|---|---|---|---|
| NuScenes | StreamMapNet Yuan et al. (2024a) | 30.1 | 29.6 | 41.9 | 33.9 |
| | HRMapNet Zhang et al. (2024a) | 30.3 | 36.9 | 44.0 | 37.1 |
| | MapUnveiler Kim et al. (2025) | 26.5 | **43.2** | 48.7 | 39.4 |
| | AugMapNet Monninger et al. (2025) | 30.3 | 39.4 | 45.3 | 38.3 |
| | Uni-PrevPredMap | **36.1** | 28.5 | **54.4** | **39.7** |
| | Uni-PrevPredMap* | 58.2 | 32.6 | 66.7 | 52.5 |
| ArgoVerse2 (2d) | StreamMapNet Yuan et al. (2024a) | 55.9 | 56.9 | 61.4 | 57.1 |
| | HRMapNet Zhang et al. (2024a) | 58.3 | 60.1 | 66.0 | 61.5 |
| | AugMapNet Monninger et al. (2025) | 57.4 | 57.4 | 61.6 | 58.8 |
| | Uni-PrevPredMap | **64.9** | **65.2** | **70.7** | **67.0** |
| | Uni-PrevPredMap* | 75.4 | 71.2 | 78.9 | 75.2 |
| ArgoVerse2 (3d) | Uni-PrevPredMap | 63.4 | 64.1 | 70.5 | 66.0 |
| | Uni-PrevPredMap* | 74.2 | 70.7 | 79.4 | 74.8 |

### A.4 COMPARISON ON NUSCENES AND ARGOVERSE2 WITH EXTENDED PERCEPTION RANGE

Following experimental configurations from established methods Wang et al. (2024b); Kim et al. (2025); Wang et al. (2024a), the extended perception range adopts 50m front/rear and 25m lateral coverage relative to the vehicle. Evaluation employs Chamfer Distance-based average precision (AP) across three thresholds: {1.0, 1.5, 2.0}m. As evidenced in Table 11, Uni-PrevPredMap sustains superior performance under extended perception range across both temporal-prior and temporal-map-fusion-prior operational modes.

Table 11: Comparison on nuScenes and ArgoVerse2 with extended perception range.

| Dataset | Method | $AP_{div}$ | $AP_{ped}$ | $AP_{bou}$ | mAP |
|---|---|---|---|---|---|
| NuScenes | SQD-MapNet Wang et al. (2024b) | 65.5 | 67.0 | 59.5 | 64.0 |
| | MapUnveiler Kim et al. (2025) | 70.0 | 68.0 | 68.2 | 68.7 |
| | PriorMapNet Wang et al. (2024a) | 67.4 | 62.5 | 65.0 | 65.0 |
| | Uni-PrevPredMap | **73.4** | **77.2** | **70.0** | **73.5** |
| | Uni-PrevPredMap* | 87.3 | 83.0 | 77.5 | 82.6 |
| ArgoVerse2 (2d) | SQD-MapNet Wang et al. (2024b) | 54.9 | 66.9 | 56.1 | 59.3 |
| | MapUnveiler Kim et al. (2025) | 67.1 | 69.7 | 59.3 | 65.4 |
| | Uni-PrevPredMap | **68.2** | **74.8** | **63.3** | **68.8** |
| | Uni-PrevPredMap* | 84.9 | 83.0 | 77.5 | 82.6 |
| ArgoVerse2 (3d) | Uni-PrevPredMap | 66.0 | 72.4 | 59.9 | 66.1 |
| | Uni-PrevPredMap* | 82.9 | 86.1 | 79.4 | 82.8 |

### A.5 PERFORMANCE COMPARISON ON CHALLENGING SCENARIOS OF NUSCENES DATASET

Table 12 presents quantitative evaluation of corner cases across challenging scenarios including occlusion (dynamic objects within 2.5m of ego vehicle per MapUnveiler Kim et al. (2025)) and adverse weather conditions (cloudy, rainy, night scenarios following BeMapNet Qiao et al. (2023)

criteria). Compared to state-of-the-art open-source methods Mask2Map Choi et al. (2024) and MapTracker Chen et al. (2024), Uni-PrevPredMap consistently outperforms in occlusion, cloudy, and night environments despite a marginal 1 mAP decrease in rainy conditions primarily attributed to MapTracker's temporal BEV feature aggregation. Crucially, Uni-PrevPredMap demonstrates significant robustness with corrupted map inputs, achieving over 3.6 mAP improvement compared to map-absent operation.

Table 12: Performance comparison on challenging scenarios of nuScenes dataset.

| Method | mAP | $mAP_{occlusion}$ | $mAP_{cloudy}$ | $mAP_{rainy}$ | $mAP_{night}$ |
|---|---|---|---|---|---|
| Mask2Map Choi et al. (2024) | 75.4 | 77.8 | 80.2 | 63.6 | 55.1 |
| MapTracker Chen et al. (2024) | 76.1 | 73.9 | 79.3 | **67.0** | 51.9 |
| Uni-PrevPredMap | **77.0** | **79.0** | **81.3** | 66.0 | **55.4** |
| Uni-PrevPredMap* | 81.8 | 83.1 | 84.9 | 72.2 | 62.2 |

### A.6 IMPACT OF REFRESHMENT THRESHOLD

The tile-indexed 3D vectorized global map processor enables automatic updating of predicted vectors (confidence > refreshment threshold) to their corresponding geolocation tiles, eliminating complex and time-consuming post-processing operations. As shown in Table 13, threshold adjustments during training approximately induce $\pm$ 1.0 mAP variation, suggesting potential systematic fluctuations inherent in post-processing pipelines. The tile-indexed 3D vectorized global map processor allows concentrated research efforts on the unified framework while maintaining operational robustness and retrieval efficiency.

Table 13: Impact of refreshment threshold."T" and "M" denote temporal and map priors respectively.

| Refreshment Threshold | | mAP (T) | mAP (T&M) |
|---|---|---|---|
| @Training | @Inference | | |
| | 0.5 | 72.9 | 80.3 |
| 0.3 | 0.6 | 73.0 | 80.1 |
| | 0.7 | 72.8 | 80.0 |
| | 0.7 | 73.7 | 81.0 |
| 0.4 | 0.8 | 74.0 | 81.0 |
| | 0.9 | 74.0 | 80.9 |
| | 0.5 | 72.6 | 79.2 |
| 0.5 | 0.6 | 73.4 | 79.5 |
| | 0.7 | 73.2 | 79.6 |

### A.7 COMPUTATIONAL COST ANALYSIS OF TILE-INDEXED 3D VECTORIZED GLOBAL MAP PROCESSOR

The computational workflow of the tile-indexed 3D vectorized global map processor comprises three main stages: retrieval, rasterization, and refreshment. First, the retrieval module queries the global tile-based storage to fetch relevant map vectors, filters them, and transforms them into the ego vehicle's coordinate frame. These vectors are then rasterized into BEV heatmaps. Lastly, the refreshment module converts the predicted vectors back to global coordinates and stores them in the global map. As summarized in Table 14, the tile-indexing mechanism keeps the combined latency of the retrieval and refreshment stages low (approximately 5 ms). The rasterization step constitutes the most computationally intensive part of the process. Overall, the processor introduces a manageable increase in total inference time.

### A.8 SPEED-ACCURACY TRADE-OFF COMPARISION WITH SOTA METHODS

Figure 6 depicts the speed-accuracy trade-offs among state-of-the-art methods, underscoring our framework's emphasis on robust, multi-mode performance. While prioritizing accuracy, our method's

Table 14: Computational cost analysis of tile-indexed 3D vectorized global map processor (measured on NVIDIA A6000). "Overall" indicates the total forward time of Uni-PrevPredMap.

| Module | Time (w/o map) | Time (w/ map) |
|---|---|---|
| Retrieval | 4.4 ms | 5.1 ms |
| Rasterization | 14.7 ms | 18.9 ms |
| Refreshment | 0.3 ms | 0.3 ms |
| Overall | 82.0 ms | 87.0 ms |

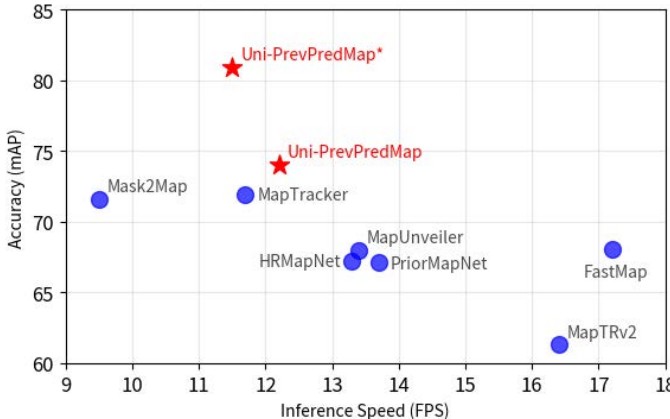

Figure 6: Speed-accuracy trade-off comparison with state-of-the-art methods.

inference speed could be further enhanced for high-speed applications, e.g., through parallelizing tile-indexed 3D vectorized global map processor, accelerating rasterization, or developing a lightweight variant.

## A.9 MORE QUALITATIVE RESULTS

Figures 7-12 demonstrate supplementary qualitative comparisons across both nuScenes and Argoverse2 datasets. Visualizations include nuScenes results under geographically non-overlapping split and extended perception range. Under all configurations, Uni-PrevPredMap[3] (temporal-map-fusion-prior mode) effectively harnesses complementary priors to iteratively refine predictions while generating enhanced temporal priors, resulting in superior forecasting performance.

We also present the global prediction maps generated through consecutive-frame reasoning on the nuScenes dataset. Figures 13-15 depict top-down comparisons between the Ground Truth and three modes of our approach: UniPrevPredMap without prior, with temporal prior only, and with both temporal and map priors. The results visually confirm that the temporal prior markedly improves temporal smoothness, while the addition of map priors further reduces both false positives and false negatives.

## A.10 BROADER IMPACT

While Uni-PrevPredMap integrates two complementary prior sources (previous predictions and corrupted HD maps) to optimize performance in online vectorized HD map construction, it does not guarantee error-free prediction of all map elements. Thus, implementing redundancy mechanisms for safety-critical applications remains essential.

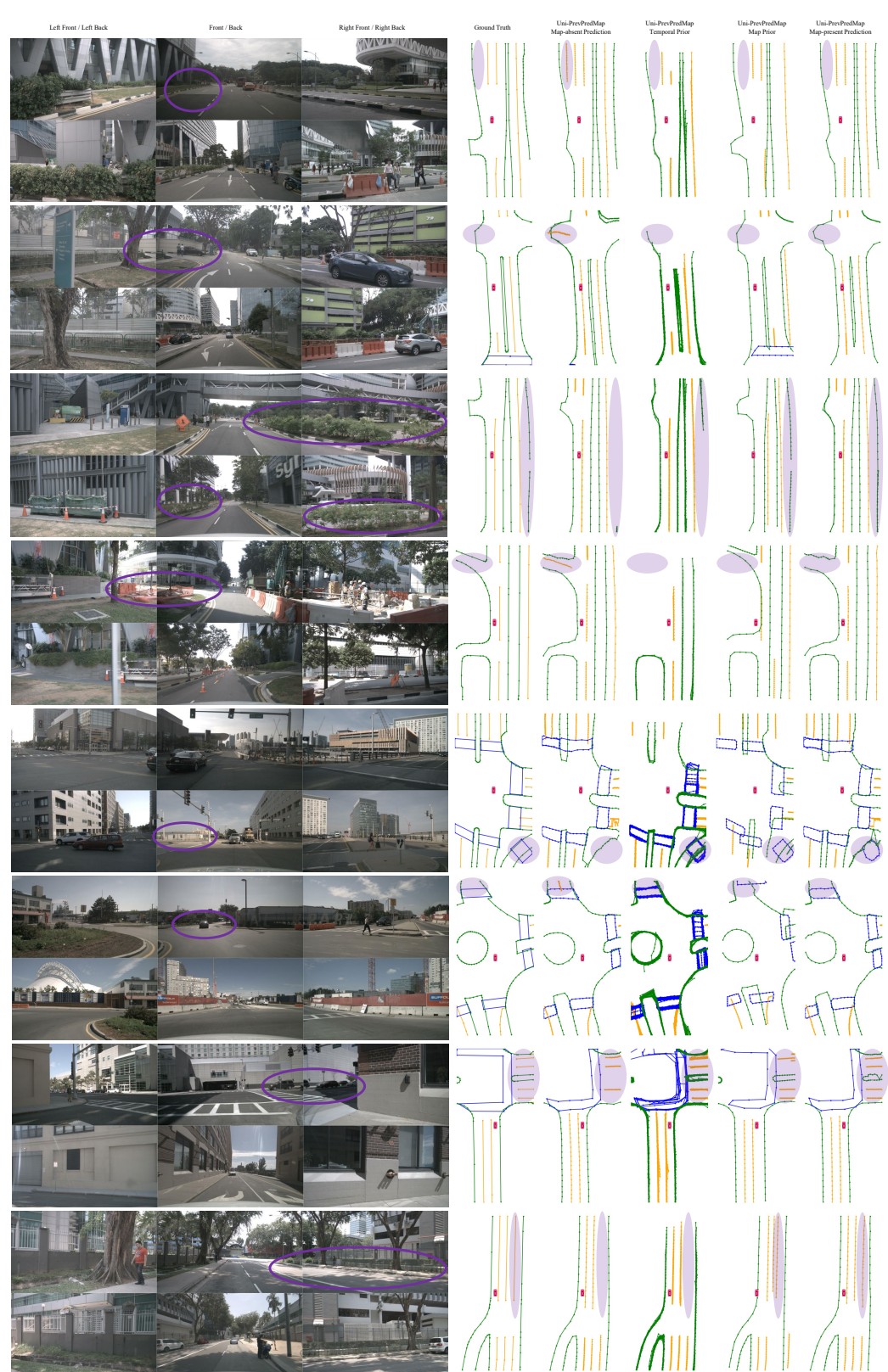

Figure 7: Additional qualitative results on the nuScenes dataset.

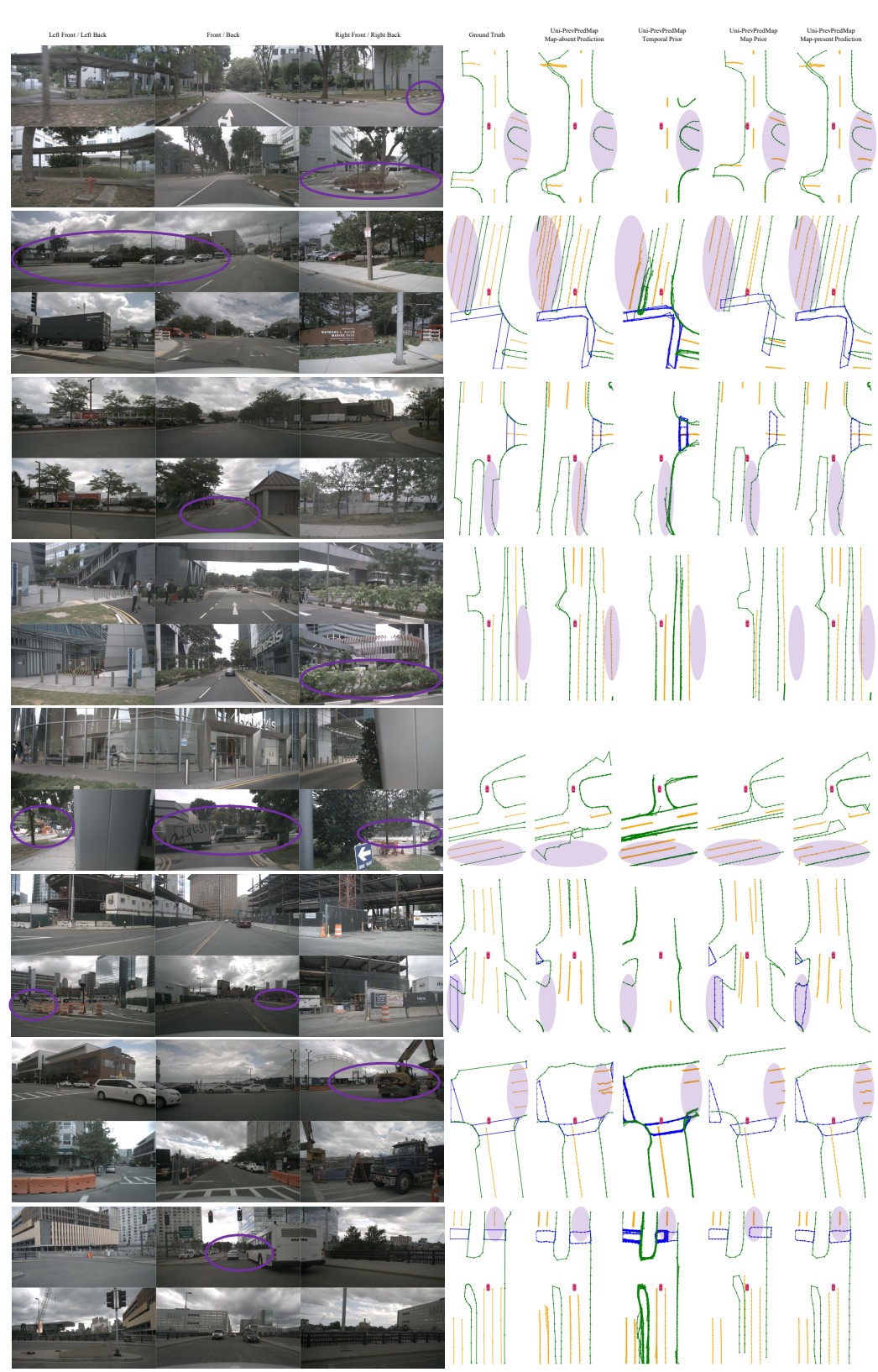

Figure 8: Additional qualitative results on the nuScenes dataset.

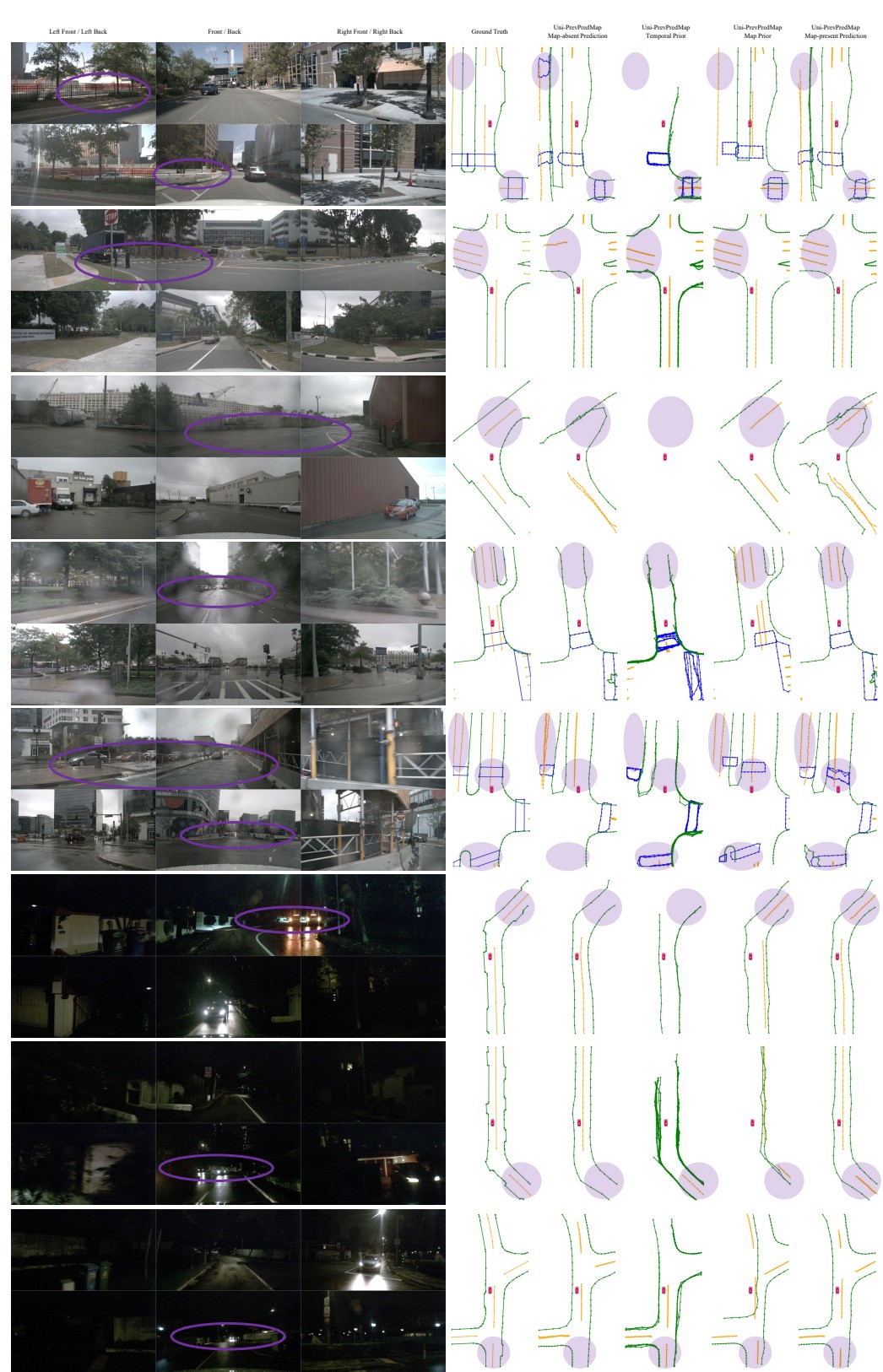

Figure 9: Additional qualitative results on the nuScenes dataset.

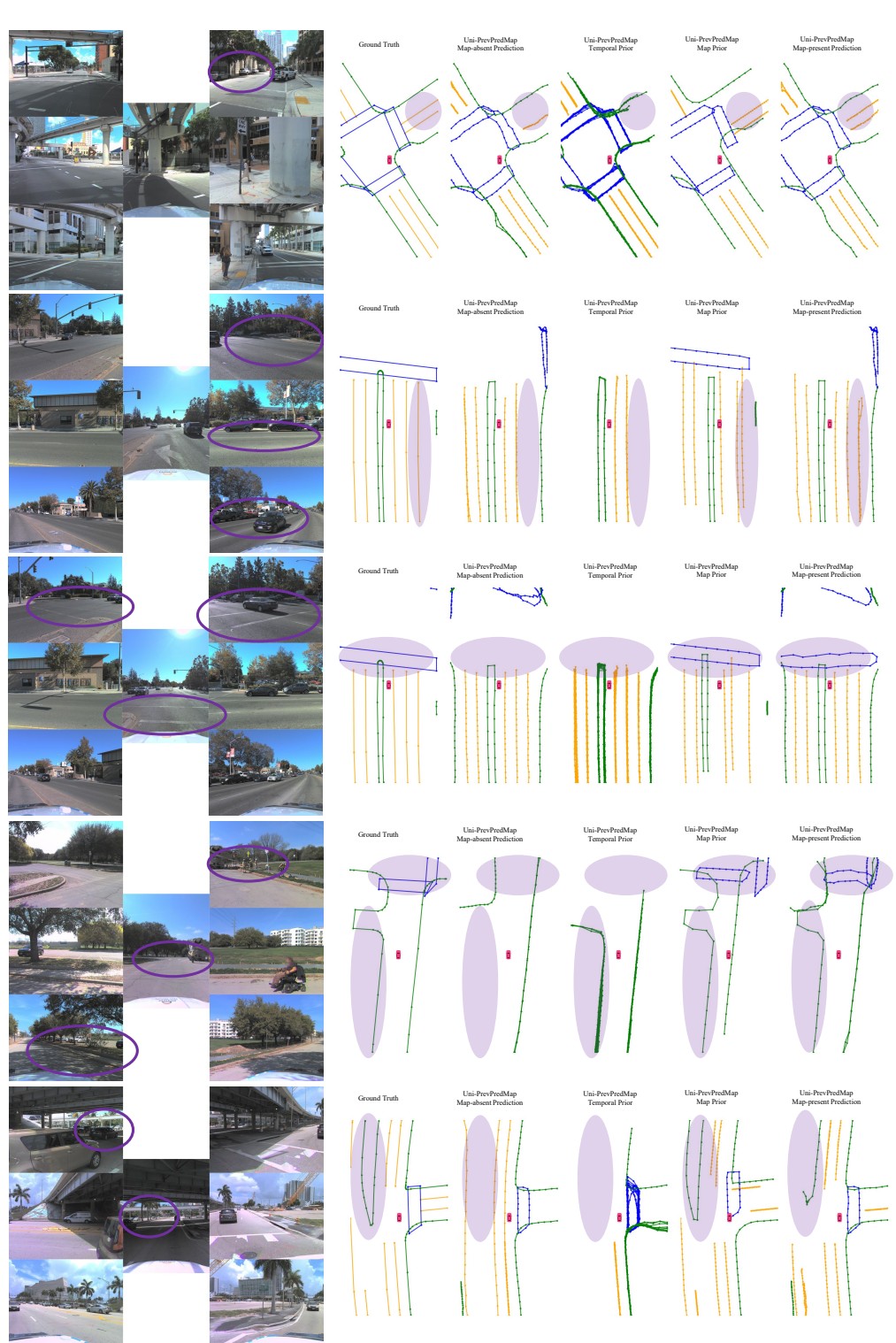

Figure 10: Additional qualitative results on the ArgoVerse2 dataset.

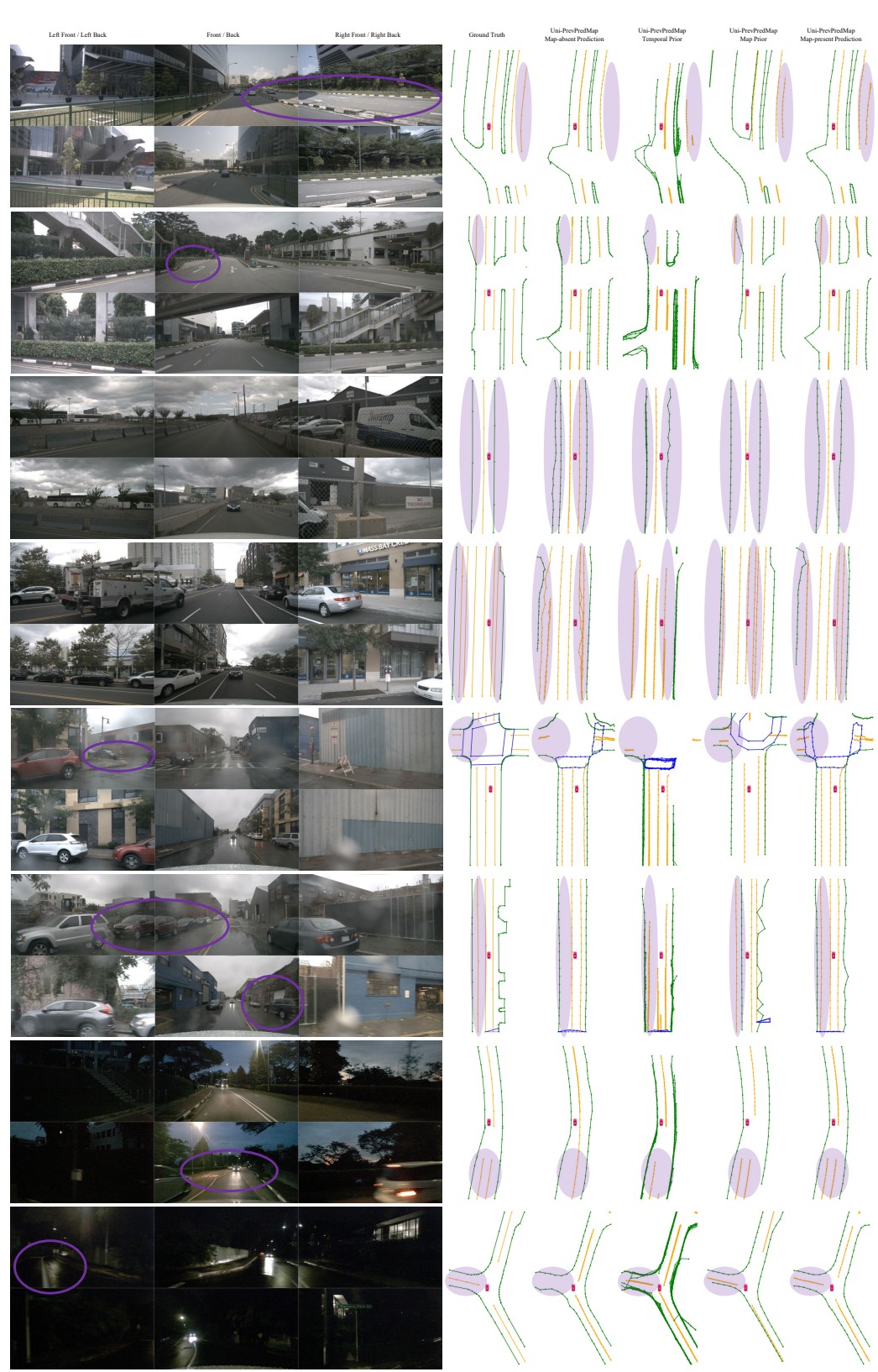

Figure 11: Additional qualitative results on the geographically non-overlapping nuScenes dataset.

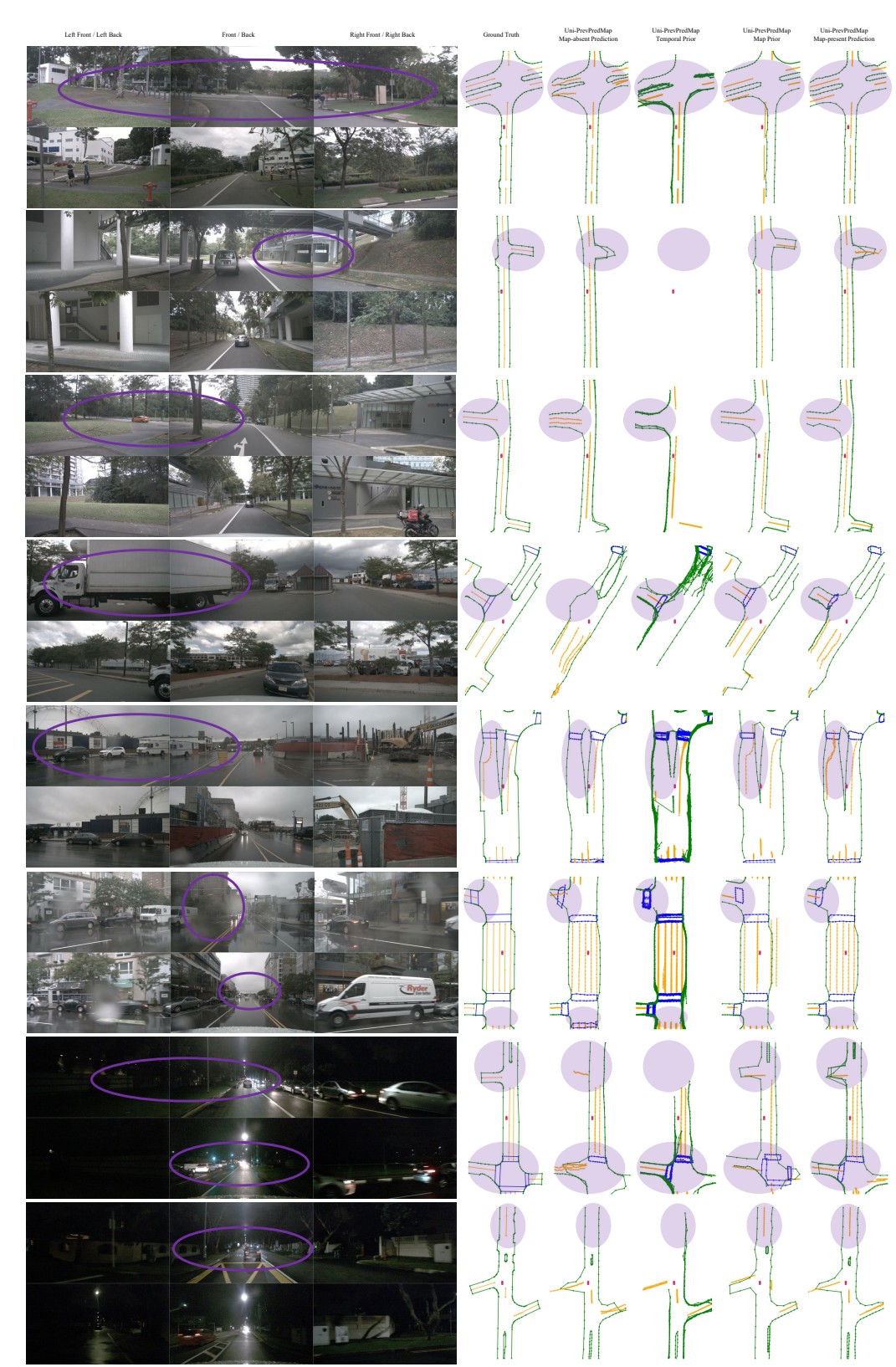

Figure 12: Additional qualitative results on the nuScenes dataset with the extended perception range.

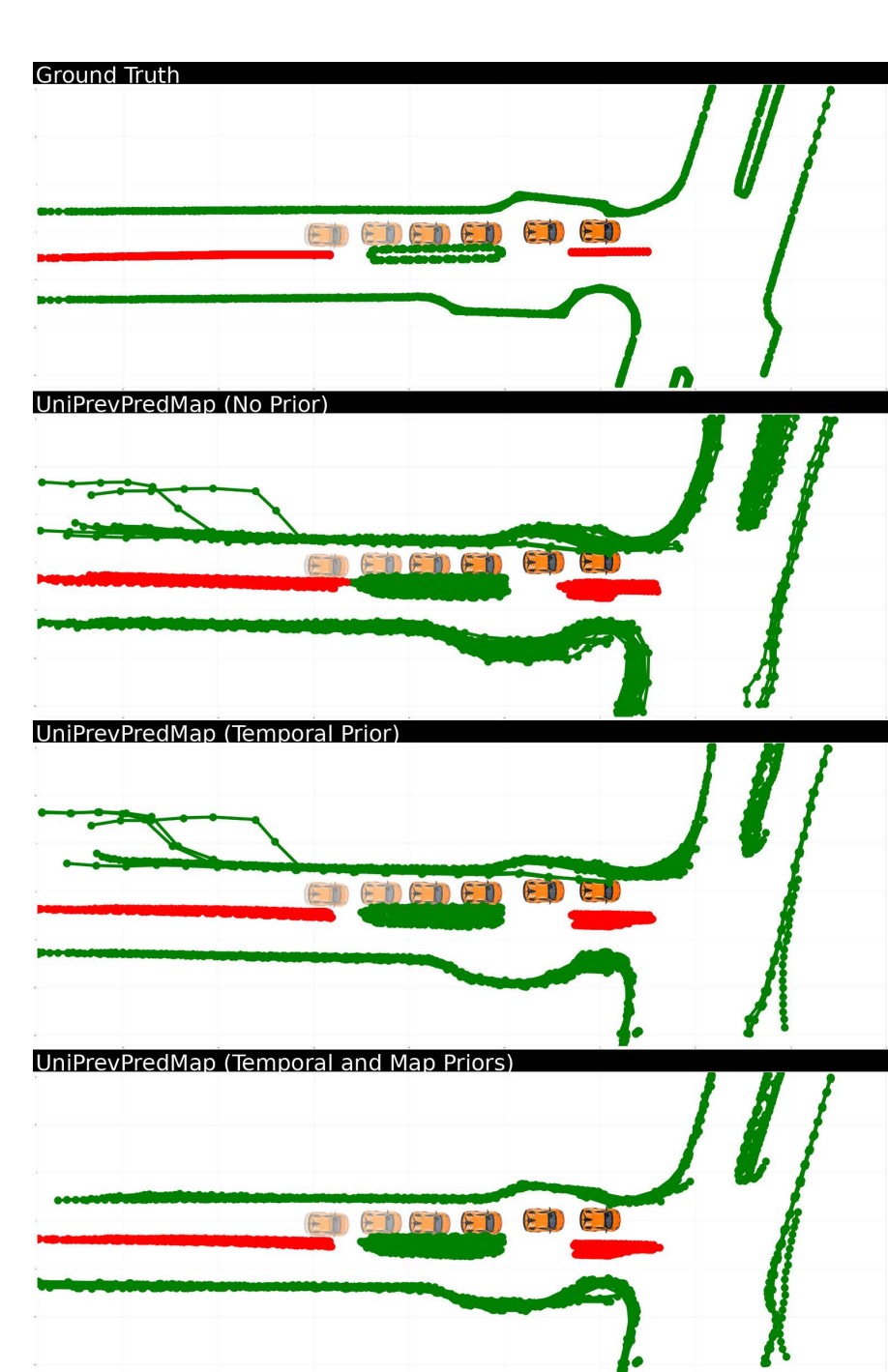

Figure 13: Additional qualitative results on the nuScenes dataset with global prediction maps.

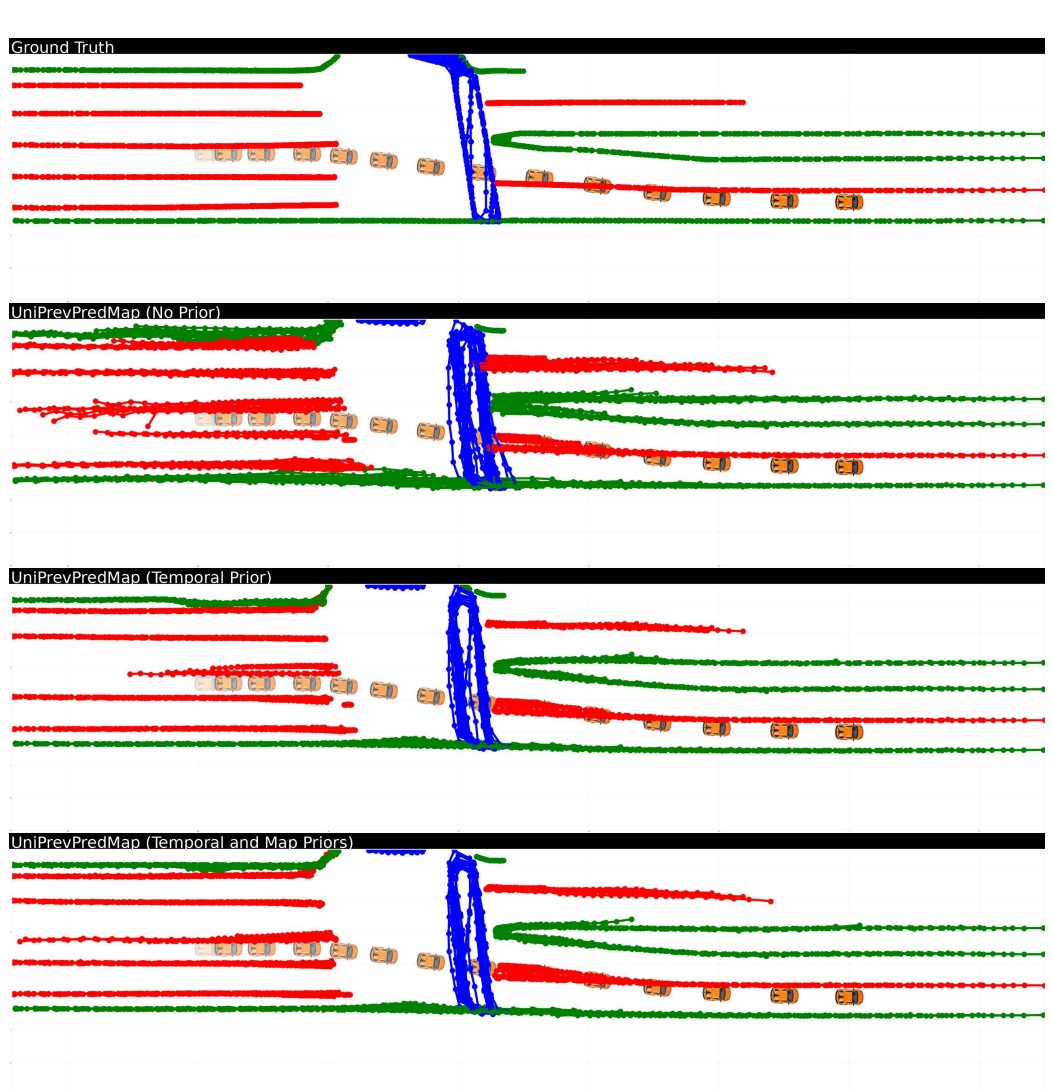

Figure 14: Additional qualitative results on the nuScenes dataset with global prediction maps.

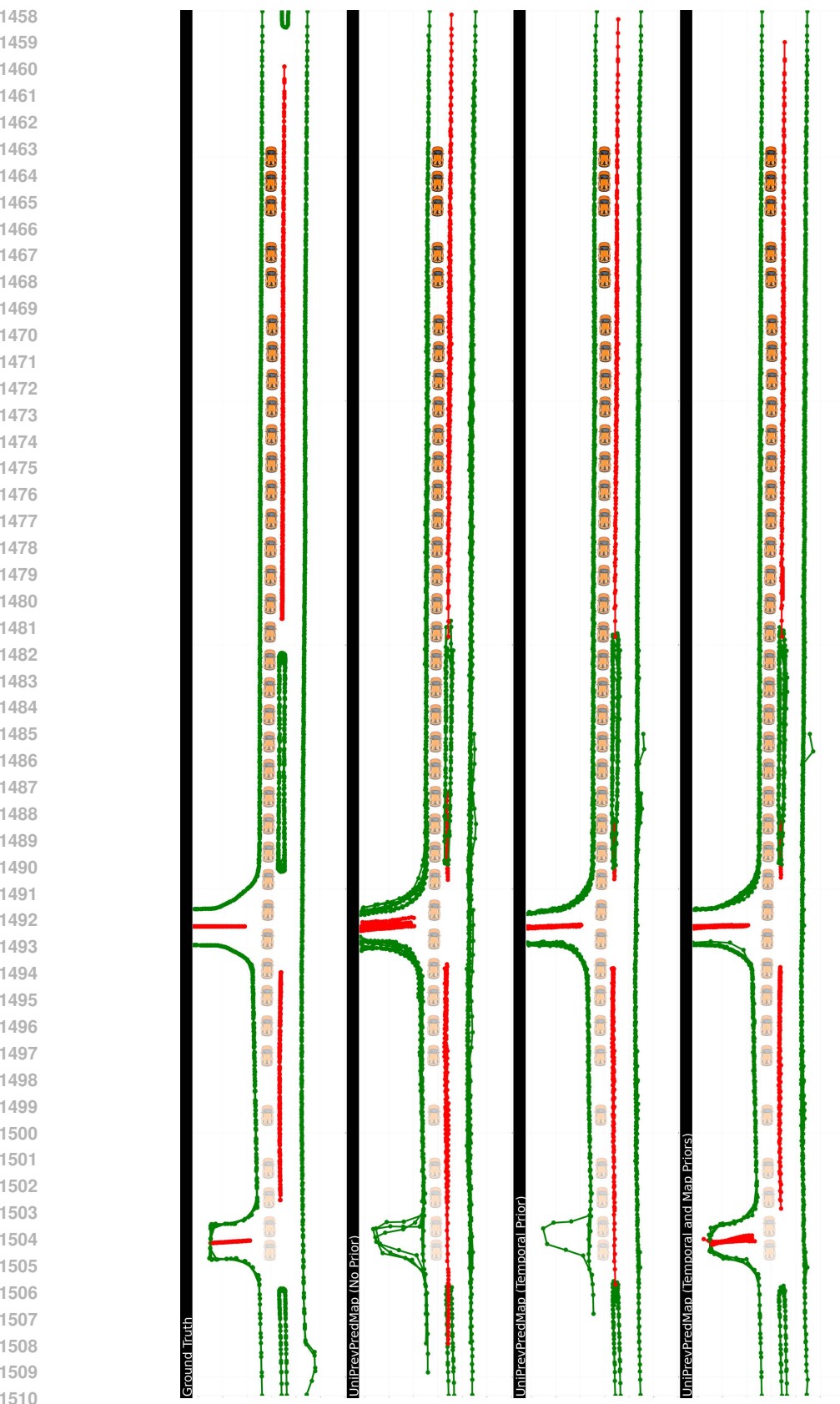

Figure 15: Additional qualitative results on the nuScenes dataset with global prediction maps.

