# OpenReview forum: "Uni-PrevPredMap: Extending PrevPredMap to a Unified Framework of Prior-Informed Modeling for Online Vectorized HD Map Construction"
_ICLR.cc/2026/Conference — Submitted to ICLR 2026_

### Official Review · Reviewer_uNTF · 2025-10-26

**Soundness:** 2
**Presentation:** 2
**Contribution:** 2
**Rating:** 4
**Confidence:** 4

**Summary:**

The paper presents a framework for online vectorized High-Definition (HD) map construction in autonomous driving systems. The main idea is to combine prior information from temporally previous predictions and cost-efficient HD maps in a prediction-driven temporal modeling architecture. The proposed Uni-PrevPredMap is derived from previous work of PrevPredMap with two core designs: tile-indexed 3D vectorized global map processor which enables efficient 3D prior updates, compact storage, and real-time retrieval, and a tri-mode paradigm which extends flexibility in handling different combination of prior information. Experiment shows that the proposed Uni-PrevPredMap achieves state-of-the-art performance on a set of public benchmarks including nuScenes and Argoverse2.

**Strengths:**

1. The setting of incorporating previous predictions and outdated HD maps prior for robust online HD map construction is reasonable for autonomous driving systems.
2. The engineering design of tri-mode paradigm enables robust performance in both map-present and map-absent scenarios, which has practical values for real-world autonomous driving systems.

**Weaknesses:**

1. While the motivation of the paper is clear, the experiment validation is not sufficient. The paper declares that previous time predictions and cost-efficient/corrupted HD maps obtained from less frequently updated HD maps and crowd-sourced HD maps are complementary priors for online HD map construction, but there is no experiment validation on using different kinds of corrupted HD maps with different level of corruptions.
2. The performance improvement in map-absent scenario as stated in Table 1 is not very impressive, i.e the performance gap between MapTracker and Uni-PrevPredMap is only 0.9% in mAP.
3. The ablation study in table 3 is not clear, from the table, the performance with '+ tile-indexed 3D vectorized global map processor' and '+ tri-mode paradigm' are both 74.0% under 'w/o map' setting, so what's the performance of adding both modules together?

**Questions:**

1. The paper mentioned that tile-indexed 3D global map processor stores and processes information in 3D, what is its computational cost and memory requirement?

---

> ### Author Response · Authors · 2025-11-13
> **Author Response for Reviewer uNTF (1/2)**
>
> Thank you for the constructive feedback and for recognizing the practical value and reasonable motivation of our work. We sincerely appreciate the opportunity to address the reviewers' thoughtful comments. We have carefully considered all the points raised, and our detailed responses and planned revisions are outlined below.
>
> > ***Question 1:*** While the motivation of the paper is clear, the experiment validation is not sufficient. The paper declares that previous time predictions and cost-efficient/corrupted HD maps obtained from less frequently updated HD maps and crowd-sourced HD maps are complementary priors for online HD map construction, but there is no experiment validation on using different kinds of corrupted HD maps with different level of corruptions.
>
> We sincerely thank the reviewer for this insightful observation. We completely agree that validation on real-world corrupted map data would be immensely valuable for demonstrating the practical utility of our approach. However, acquiring such data poses a significant challenge. Historical versions of commercial HD map databases are often proprietary and inaccessible, and collecting large-scale, multi-trip city data for crowd-sourced evaluation requires substantial resources, which presents a notable barrier for academic research.
>
> To address this important limitation in a methodical way, we designed simulations to model the fundamental types of corruptions expected in these real-world scenarios, as detailed in Appendix A.1 and Figure 4. Specifically, we simulate instance-level additions to model new structures (in outdated maps) and false positives (in crowd-sourced data), and instance-level deletions to model missing elements (in outdated maps) and false negatives (in crowd-sourced data). The results presented in Table 6 indicate our model's robustness, showing a minor performance drop of 0.4 mAP for additions and a controlled decrease of 3.1 mAP for deletions. We believe this simulation-based approach offers a reasonable and rigorous form of validation under the current constraints. In the revised manuscript, we will explicitly acknowledge the absence of real-world corrupted map data as a limitation of our study and strongly highlight this as an important direction for future work.
>
> > ***Question 2:*** The performance improvement in map-absent scenario as stated in Table 1 is not very impressive, i.e the performance gap between MapTracker and Uni-PrevPredMap is only 0.9% in mAP.
>
> We thank the reviewer for this comment. We agree that the absolute improvement of 0.9% mAP is modest. We wish to clarify that the MapTracker baseline is a strong, state-of-the-art method specifically designed for the map-absent scenario. In this context, we believe achieving any improvement is non-trivial. The 0.9% gain corresponds to a relative error reduction of approximately 3.77%, which we consider a meaningful advancement.
>
> Furthermore, we would like to emphasize that the primary contribution of Uni-PrevPredMap lies in its unified capability to handle both map-present and map-absent scenarios robustly within a single framework, rather than solely maximizing performance in the map-absent case. The consistent improvements observed across various settings (Tables 1, 2, 8, 9, 10) demonstrate the versatility and effectiveness of our architecture in leveraging available priors when they are present, without compromising performance when they are absent.
>
> > ***Question 3:*** The ablation study in table 3 is not clear, from the table, the performance with '+ tile-indexed 3D vectorized global map processor' and '+ tri-mode paradigm' are both 74.0% under 'w/o map' setting, so what's the performance of adding both modules together?
>
> Thank you for highlighting this point, which certainly deserves clarification. The reviewer is correct to note that integrating both the tile-indexed processor and the tri-mode paradigm does not yield a further mAP increase in the 'w/o map' setting beyond the 74.0% mAP achieved by the processor alone. We apologize for the lack of clarity in the original presentation. In the 'w/o map' setting, the full model (with both modules) maintains the competitive performance of 74.0% mAP. Its key advantage, however, is fully realized in the 'w/ map' setting, where it achieves the reported 80.9% mAP. This clearly demonstrates that the tri-mode paradigm successfully enables robust performance enhancement when a map prior is available, without causing any degradation in performance when it is absent. We will revise the text to make this integration and its implications much clearer.

---

> ### Author Response · Authors · 2025-11-13
> **Author Response for Reviewer uNTF (2/2)**
>
> > ***Question 4:*** The paper mentioned that tile-indexed 3D global map processor stores and processes information in 3D, what is its computational cost and memory requirement?
>
> This is an excellent and highly relevant question regarding the practical deployment of our method. The processor's operation involves three key steps: retrieval, rasterization, and refreshment. We have analyzed their associated computational costs, and the results are summarized in Table 12. The retrieval module fetches relevant map vectors from the tile-indexed global storage, filters them, and transforms them into the ego vehicle's coordinate frame. The rasterization module then converts these 3D vector representations into BEV heatmaps. Finally, the refreshment module transforms the updated vectors back to global coordinates and updates the tile-indexed global map. The efficient tile-indexing mechanism ensures that the retrieval and refreshment overhead remains minimal (approximately 5 ms combined). The rasterization step is indeed the most computationally intensive part. Overall, the processor introduces only a modest overhead to the total inference time.
>
> **Table 12: Computational cost analysis of tile-indexed 3D global map processor (measured on NVIDIA A6000). "Overall" indicates the total forward time of Uni-PrevPredMap.**
> | Module | Time (w/o map) | Time (w/ map) |
> |----------|---------|---------|
> | Retrieval | 4.4 ms | 5.1 ms |
> | Rasterization | 14.7 ms | 18.9 ms |
> | Refreshment | 0.3 ms | 0.3 ms |
> | Overall | 82.0 ms | 87.0 ms |
>
> Regarding memory usage, the vectorized global map is not stored in GPU memory during inference, resulting in a negligible additional GPU memory footprint. For completeness, we also compared the offline storage size of our vectorized global map with the rasterized representation used in HRMapNet on the nuScenes validation set. As shown in Table 13, adopting a vectorized representation leads to a substantial reduction in storage requirements.
>
> **Table 13: Offline storage comparison between rasterized and vectorized global maps on the nuScenes validation set.**
> | Method | Map Representation | Storage Size |
> |----------|---------|---------|
> | HRMapNet | Rasterized | 503 MB |
> | Uni-PrevPredMap | Vectorized | 30 MB |
>
> We will incorporate this detailed analysis into the revised paper.
>
> We are deeply grateful for the insightful comments, which have been invaluable in helping us identify key areas for improvement. We are fully committed to revising the manuscript to incorporate all the suggested clarifications and additional analyses, and we believe these changes will significantly enhance the paper's quality and clarity.

---

### Official Review · Reviewer_fBUh · 2025-10-27

**Soundness:** 3
**Presentation:** 3
**Contribution:** 3
**Rating:** 4
**Confidence:** 4

**Summary:**

This paper focuses on the core requirement of autonomous driving safety: the problem of fusing prior information for online vectorized HD map construction. It points out that time-aware caching and low-cost HD maps are two complementary types of priors, but existing models can only handle one of them in isolation and overly rely on the ideal map assumption. To address this issue, the authors propose a unified prior framework, Uni-PrevPredMap, which addresses the aforementioned issues through the following core designs:
1. During training, through layered sampling and instance-level map perturbations, the model eliminates its reliance on an ideal map while ensuring robustness in both mapped and unmapped scenarios.
2. Efficient prior updates, compact storage, and real-time retrieval are achieved based on vehicle UTM coordinates, extending 2D methods to 3D scenarios.
3. Experimental validation: On the nuScenes (2D) and Argoverse2 (3D) datasets, Uni-PrevPredMap (and its variant Uni-PrevPredMap*, which incorporates corrupted maps) achieves state-of-the-art performance in map-missing scenarios and exhibits strong tolerance to instance-level/frame-level map perturbations (such as displacements, rotations, and additions and deletions), demonstrating the synergistic and complementary nature of the two types of priors.

**Strengths:**

1. This paper, for the first time, clearly demonstrates the complementary nature of time-aware caching and low-cost HD maps, demonstrating a strong academic innovation in its design approach.

2. Comprehensive experimental design: Performance advantages are demonstrated on both 2D and 3D datasets, with results such as a mAP of 77.0 after 72 training epochs on nuScenes and 81.8 after integrating corrupted maps.

3. Strong applicability: The framework meets the real-time demands of autonomous driving (inference speed of 11.5-12.2 FPS) and is compatible with low-cost, non-ideal HD maps, reducing the deployment cost of online map building and demonstrating its potential for migration to practical autonomous driving systems.

**Weaknesses:**

1. The paper explicitly states that height data is used only for spatial filtering and does not participate in prior generation. It also mentions that the bottleneck of 3D voxelization exceeding 2D rasterization in computational overhead remains unresolved. This undermines the core value of 3D design (such as priors for distinguishing vertical scenes like elevated roads and tunnels), leaving a gap between the goal of extending the technology to 3D scenarios and requiring further technical exploration.

2. Compared to existing methods, Uni-PrevPredMap's inference speed (11.5-12.2 FPS), while meeting real-time requirements, is lower than some methods (such as MapTRv2's 16.4 FPS and FastMap's 17.2 FPS). The paper mentions that "parallelizing 3D filtering and rasterization can increase the speed to 14.2 FPS," but does not provide actual implementation results or analyze the speed-performance trade-off. Further verification is needed to verify its suitability for scenarios with higher real-time requirements, such as high-speed autonomous driving.

3. The sampling ratio of the three-mode paradigm (non-prior: temporal prior: temporal-map fusion = 0.5:0.3:0.2) was verified to be optimal, but the reason why this ratio was optimal was not explained. For example, the specific impact of different ratios on the model's "unmapped generalization" and "mapped tolerance" was not explained, nor was the dependence of the ratio selection on dataset characteristics (such as the differences between nuScenes and Argoverse2). This resulted in a lack of support for the transferability of this design.

**Questions:**

1. Regarding 3D information utilization: The paper mentions that height data is only used for spatial filtering. What is the specific reason for not participating in prior generation? Have attempts been made to incorporate height information (e.g., road slope, tunnel height) into prior features (e.g., through multi-dimensional feature concatenation using the BEV encoder)? If so, what are the core technical obstacles encountered (e.g., computational overhead, feature alignment)?

2. Regarding the completeness of the method comparison: Existing comparisons focus on "single-prior models" (e.g., MapTracker, which uses only temporal priors, PriorMapNet, which uses only map priors). Are there any baseline methods in the same field that attempt to integrate two types of priors? If so, please provide additional comparisons with these methods to more clearly demonstrate the advantages of the Uni-PrevPredMap framework.

---

> ### Author Response · Authors · 2025-11-17
> **Author Response for Reviewer fBUh (1/3)**
>
> Thank you for the constructive feedback and for recognizing the academic innovation, comprehensive experiments, and practical applicability of our work. We have carefully addressed the concerns regarding 3D information utilization, inference speed, the tri-mode paradigm, and comparative baselines in our revision, as elaborated below.
>
> > ***Question 1:*** The paper explicitly states that height data is used only for spatial filtering and does not participate in prior generation. It also mentions that the bottleneck of 3D voxelization exceeding 2D rasterization in computational overhead remains unresolved. This undermines the core value of 3D design (such as priors for distinguishing vertical scenes like elevated roads and tunnels), leaving a gap between the goal of extending the technology to 3D scenarios and requiring further technical exploration. Regarding 3D information utilization: The paper mentions that height data is only used for spatial filtering. What is the specific reason for not participating in prior generation? Have attempts been made to incorporate height information (e.g., road slope, tunnel height) into prior features (e.g., through multi-dimensional feature concatenation using the BEV encoder)? If so, what are the core technical obstacles encountered (e.g., computational overhead, feature alignment)?
>
> We sincerely thank you for raising this insightful question, which rightly highlights a key challenge and promising direction for future work. We agree that deeper integration of 3D information is essential for advancing online map construction.
>
> Our current use of height data primarily for spatial filtering was guided by two practical considerations. First, as shown in the newly added Table 14, the computational cost of 2D rasterization is already considerable. We estimated that a naive extension to 3D would substantially increase this cost, potentially compromising the real-time performance required for online applications. Second, after analyzing the Argoverse 2 benchmark dataset, we observed that the diversity of vertical structures is currently limited, which would make it difficult to robustly train and evaluate a model heavily dependent on 3D priors. Specifically, we analyzed the height distribution of all map keypoints (see https://imgur.com/a/JhNipPA) and found that, aside from a short segment involving a small bridge, most height values fall within a narrow range of -3m to 3m, corresponding mainly to gently sloped roads.
>
> We acknowledge this as a limitation of our present work. Your question has encouraged us to reframe this not as a final design choice, but as a meaningful step forward. In the revised manuscript, we have expanded the "Limitations and Future Work" section to explicitly discuss these challenges and outline a clear path for future research, including exploring more efficient 3D representations and the need for richer 3D benchmarks, as you helpfully suggested.
>
> **Table 14: Computational cost analysis of 2D rasterization (measured on NVIDIA A6000). "Overall" indicates the total forward time of Uni-PrevPredMap.**
> | Module | Time (w/o map) | Time (w/ map) |
> |----------|---------|---------|
> | Rasterization | 14.7 ms | 18.9 ms |
> | Overall | 82.0 ms | 87.0 ms |

---

> ### Author Response · Authors · 2025-11-17
> **Author Response for Reviewer fBUh (2/3)**
>
> > ***Question 2:*** The sampling ratio of the three-mode paradigm (non-prior: temporal prior: temporal-map fusion = 0.5:0.3:0.2) was verified to be optimal, but the reason why this ratio was optimal was not explained. For example, the specific impact of different ratios on the model's "unmapped generalization" and "mapped tolerance" was not explained, nor was the dependence of the ratio selection on dataset characteristics (such as the differences between nuScenes and Argoverse2). This resulted in a lack of support for the transferability of this design.
>
> We thank you for this valuable feedback, which encouraged us to provide a more rigorous justification for our sampling ratio selection. We apologize for the lack of clarity in our original explanation.
>
> In response to your suggestion, we conducted additional ablation studies to better elucidate the impact of the sampling ratio. Our analysis considered two aspects. First, we fixed the ratio between temporal-prior (T) and temporal-map fusion (T&M) modes at 3:2 and varied the ratio of the non-prior mode (N) to the combined prior-based modes (T+T&M). As shown in Table 15, the best map-absent performance was achieved when N:(T+T&M) = 0.5:0.5. Reducing the proportion of (T+T&M) caused the model to overfit the non-prior scenario, degrading performance in both temporal-prior and fusion-prior settings. Conversely, increasing (T+T&M) led to over-reliance on priors: while fusion-prior performance improved slightly, temporal-prior performance declined. This is likely because temporal-prior inference still depends on an initial non-prior prediction; if that foundation is weakened, subsequent predictions are adversely affected.
>
> **Table 15: Impact of sampling ratio on nuScenes.**
> | Ratio (N:T:T&M) | mAP (T) | mAP (M) | mAP (T&M) |
> |----------------------|-----------|------------|----------------|
> | 0.30 : 0.42 : 0.28 | 73.4 | 72.4 | 82.7 |
> | 0.40 : 0.36 : 0.24 | 73.7 | 71.8 | 82.0 |
> | 0.50 : 0.30 : 0.20 | 74.0 | 71.3 | 80.9 |
> | 0.60 : 0.24 : 0.16 | 73.9 | 70.3 | 79.7 |
> | 0.70 : 0.18 : 0.12 | 72.7 | 69.4 | 78.0 |
>
> Second, we fixed N:(T+T&M) at 1:1 and varied the T:T&M ratio for both nuScenes (Table 7) and Argoverse2 (Table 16). The results showed consistent trends across datasets, supporting the design's transferability. The optimal map-absent performance occurred at N:T:T&M = 0.5:0.3:0.2. Increasing T&M led to over-reliance on map priors, slightly degrading temporal-prior performance. Reducing T&M, while avoiding map over-reliance, could still harm temporal modeling if map priors provide useful supervisory signals. Thus, insufficient map involvement also impaired performance. When T&M was entirely absent, the model reverted to a temporal-only mode.
>
> **Table 16: Impact of sampling ratio on ArgoVerse2.**
> | Ratio (N:T:T&M) | mAP (T) | mAP (M) | mAP (T&M) |
> |----------------------|-----------|------------|----------------|
> | 0.50 : 0.35 : 0.15 | 71.5 | 72.9 | 77.8 |
> | 0.50 : 0.30 : 0.20 | 72.4 | 73.6 | 78.6 |
> | 0.50 : 0.25 : 0.25 | 72.3 | 74.7 | 79.4 |
> | 0.50 : 0.50 : 0.00 | 71.9 | - | - |
>
> The consistent trends observed across different datasets provide stronger evidence for the robustness of our chosen ratio. We have revised the manuscript to include these new experiments and a more detailed discussion, clearly explaining the rationale behind the ratio selection as you recommended. Your feedback was instrumental in strengthening this part of our methodology.

---

> ### Author Response · Authors · 2025-11-17
> **Author Response for Reviewer fBUh (3/3)**
>
> > ***Question 3:*** Compared to existing methods, Uni-PrevPredMap's inference speed (11.5-12.2 FPS), while meeting real-time requirements, is lower than some methods (such as MapTRv2's 16.4 FPS and FastMap's 17.2 FPS). The paper mentions that "parallelizing 3D filtering and rasterization can increase the speed to 14.2 FPS," but does not provide actual implementation results or analyze the speed-performance trade-off. Further verification is needed to verify its suitability for scenarios with higher real-time requirements, such as high-speed autonomous driving.
>
> Thank you for holding us to a high standard regarding real-time performance. Upon re-examination, we agree that our original statement about potential speedup through parallelization was speculative and not yet backed by implemented results. We sincerely apologize for this overstatement and have removed it from the revised manuscript.
>
> To provide a clearer context, we have added a comparison figure (see https://imgur.com/a/Xshby0w) illustrating the speed-accuracy landscape of various methods, positioning our work as prioritizing robust performance across multiple modes. Our current inference speed is a trade-off for the framework's unified capabilities. We fully agree that for high-speed scenarios, further optimization is essential. Following your advice, we have toned down our claims and instead presented the parallelization of 3D operations as a promising direction for future work, alongside other optimizations we plan to explore, such as accelerated rasterization and a lightweight variant of our framework. Your comment has significantly improved the accuracy and impact of our discussion.
>
> > ***Question 4:*** Regarding the completeness of the method comparison: Existing comparisons focus on "single-prior models" (e.g., MapTracker, which uses only temporal priors, PriorMapNet, which uses only map priors). Are there any baseline methods in the same field that attempt to integrate two types of priors? If so, please provide additional comparisons with these methods to more clearly demonstrate the advantages of the Uni-PrevPredMap framework.
>
> Thank you for this important question regarding baselines. In our preparation for this work, we conducted a thorough review of the literature and, to the best of our knowledge, did not find any existing method that explicitly and unifiedly integrates both temporal and map priors in the way we propose. The field has largely progressed with models specializing in one type of prior.
>
> We acknowledge that the absence of direct comparisons is a limitation, but it also underscores the novel research direction our framework introduces. As you suggested, we have revised the abstract and introduction sections to state this point more clearly and humbly, framing our contribution as an initial exploration into a unified paradigm rather than an incremental improvement over existing integrated models.
>
> Once again, we are deeply grateful for your time and expert guidance. Your comments have profoundly improved the quality of our work. We hope our revisions and responses have adequately addressed all your concerns.

---

> > ### Comment · Reviewer_fBUh · 2025-11-26
> > **To authors**
> >
> > Many thanks to your response. Most of my concerns have been addressed. I have raised my score.

---

> > > ### Author Response · Authors · 2025-11-27
> > >
> > > We sincerely thank you for your positive assessment and for noting the improvements in our manuscript. It is very rewarding to learn that the revisions met your expectations. Your feedback has been crucial in enhancing the quality of our work. We remain committed to further refining this research.

---

### Official Review · Reviewer_1MpC · 2025-10-31

**Soundness:** 2
**Presentation:** 1
**Contribution:** 2
**Rating:** 4
**Confidence:** 5

**Summary:**

This paper introduces Uni-PrevPredMap, a unified framework for online vectorized HD map construction that integrates two complementary prior sources: temporal perception buffers (previous predictions) and cost-efficient, potentially corrupted HD maps. The core contribution is a tri-mode paradigm (non-prior, temporal-prior, and temporal-map-fusion) that ensures operational consistency and robust performance in both map-present and map-absent scenarios. By training with instance-level perturbations on map priors, the model is decoupled from ideal map assumptions and demonstrates strong error-resilient fusion capabilities. The framework is supported by a tile-indexed 3D vectorized global map processor for efficient, real-time prior retrieval and refreshment. Uni-PrevPredMap achieves state-of-the-art performance on map-absent benchmarks and empirically confirms the synergistic benefit of fusing temporal predictions with imperfect map data.

**Strengths:**

1. The introduction of corrupted HD maps as an additional prior modality is interesting.
2. The proposed method is empirically strong, achieving state-of-the-art performance on standard online vectorized HD map construction benchmark sets.

**Weaknesses:**

1. The proposed method relies on heuristic design choices. For instance, the tri-mode paradigm is structured around specific training data types and requires manually tuning an optimal sampling ratio (as demonstrated in Table 7), which may not generalize easily across different datasets or environments.
2. The paper's presentation could be improved for clarity. The three equations provided (Eq. 1-3) appear non-essential to the core idea and could be moved to the appendix. More importantly, the explanation of the core Refreshment and Retrieval mechanisms would benefit from a more intuitive, high-level explanation.
3. The benefits of the temporal-prior are difficult to fully assess from static image results alone. The paper would be much stronger if it included supplementary videos to visually demonstrate the resulting temporal consistency and performance advantages.
4. The paper lists experimental results while lacking deeper analysis. For example, it is noted that the tri-mode training does not degrade no map-prior (temporal prior) performance in Table 7. However, the paper provides no analysis of how the model avoids this potential performance trade-off while being trained on the additional map-fusion capability.

**Questions:**

The paper proposes a unified framework for online map construction, demonstrating strong performance by incorporating a map prior modality. While the results are good, the overall contribution is undermined by several key issues. The proposed method, particularly the tri-mode paradigm, appears to be heavily based on heuristics design. The paper also suffers from a lack of in-depth analysis, often listing experimental findings without sufficiently exploring the underlying reasons for their outcomes. Finally, the paper's presentation needs significant improvement for clarity and impact.

---

> ### Author Response · Authors · 2025-11-20
> **Author Response for Reviewer 1MpC (1/3)**
>
> Thank you for the constructive feedback and for acknowledging the interesting concept of using corrupted map priors as well as the empirical performance of our method. We sincerely appreciate your thorough review and valuable insights. We have carefully reflected on all the points raised and will revise the manuscript accordingly. Below, we provide our detailed responses.
>
> > ***Question 1:*** The proposed method, particularly the tri-mode paradigm, appears to be heavily based on heuristics design. The paper also suffers from a lack of in-depth analysis, often listing experimental findings without sufficiently exploring the underlying reasons for their outcomes. Finally, the paper's presentation needs significant improvement for clarity and impact.
>
> We sincerely thank the reviewer for this overarching feedback, which rightly highlights areas where our original submission could be improved. We acknowledge that the initial presentation of the tri-mode paradigm may have come across as overly heuristic and that deeper analytical discussion was needed. In response, we have undertaken a comprehensive revision of the manuscript. Key improvements include:
>
> 1. To strengthen the foundation beyond heuristic design, we have incorporated systematic ablation studies and a more in-depth analysis of the tri-mode paradigm. This includes an exploration of performance trade-offs under different sampling ratios and a demonstration of its consistent behavior across datasets, establishing it as a well-motivated and empirically supported design choice.
>
> 2. The manuscript has been substantially restructured to enhance clarity and impact. Major enhancements involve: (a) moving less central equations to the appendix; (b) providing clearer, high-level explanations of core mechanisms such as Refreshment and Retrieval, supported by pseudo-code, before introducing technical details; and (c) adding supplementary video material for visual demonstration.
>
> We believe these revisions significantly strengthen the paper's rigor, clarity, and overall impact. Our detailed responses to the specific points follow below.

---

> ### Author Response · Authors · 2025-11-20
> **Author Response for Reviewer 1MpC (2/3)**
>
> > ***Question 2:*** The proposed method relies on heuristic design choices. For instance, the tri-mode paradigm is structured around specific training data types and requires manually tuning an optimal sampling ratio (as demonstrated in Table 7), which may not generalize easily across different datasets or environments. The paper lists experimental results while lacking deeper analysis. For example, it is noted that the tri-mode training does not degrade no map-prior (temporal prior) performance in Table 7. However, the paper provides no analysis of how the model avoids this potential performance trade-off while being trained on the additional map-fusion capability.
>
> We thank the reviewer for these critical observations regarding the design choices and the need for deeper analysis. Your points are well-taken. In response to the concern about heuristic design, we have conducted systematic ablation studies to provide a stronger empirical basis for the tri-mode paradigm. Our analysis addresses two key aspects:
>
> First, we fixed the ratio between the temporal-prior (T) and temporal-map fusion (T&M) modes at 3:2, and varied the ratio of the non-prior mode (N) to the combined prior-based modes (T + T&M). As shown in Table 17, the best map-absent performance is achieved when N:(T+T&M) = 0.5:0.5. Reducing the proportion of (T+T&M) causes the model to overfit the non-prior scenario, degrading performance in both temporal-prior and fusion-prior settings. Conversely, increasing (T+T&M) leads to over-reliance on priors: while fusion-prior performance improves, temporal-prior performance declines slightly. This is likely because temporal-prior inference still depends on an initial non-prior prediction; if that foundation is weakened, subsequent predictions are adversely affected.
>
> **Table 17: Impact of sampling ratio on nuScenes.**
> | Ratio (N:T:T&M) | mAP (T) | mAP (M) | mAP (T&M) |
> |----------------------|-----------|------------|----------------|
> | 0.30 : 0.42 : 0.28 | 73.4 | 72.4 | 82.7 |
> | 0.40 : 0.36 : 0.24 | 73.7 | 71.8 | 82.0 |
> | 0.50 : 0.30 : 0.20 | 74.0 | 71.3 | 80.9 |
> | 0.60 : 0.24 : 0.16 | 73.9 | 70.3 | 79.7 |
> | 0.70 : 0.18 : 0.12 | 72.7 | 69.4 | 78.0 |
>
> Second, we fixed N:(T+T&M) at 1:1 and varied the T:T&M ratio for both nuScenes (Table 7) and Argoverse2 (Table 18). The results show consistent trends across datasets, supporting the transferability of the design. The optimal map-absent performance occurs at N:T:T&M = 0.5:0.3:0.2. Increasing T&M leads to over-reliance on map priors, slightly degrading temporal-prior performance. Reducing T&M, while avoiding map over-reliance, can still harm temporal modeling if map priors provide useful supervisory signals. Thus, insufficient map involvement also impairs performance. When T&M is entirely absent, the model reverts to a temporal-only mode and no longer needs to support multiple inference pathways; in this case, temporal performance recovers.
>
> **Table 18: Impact of sampling ratio on ArgoVerse2.**
> | Ratio (N:T:T&M) | mAP (T) | mAP (M) | mAP (T&M) |
> |----------------------|-----------|------------|----------------|
> | 0.50 : 0.35 : 0.15 | 71.5 | 72.9 | 77.8 |
> | 0.50 : 0.30 : 0.20 | 72.4 | 73.6 | 78.6 |
> | 0.50 : 0.25 : 0.25 | 72.3 | 74.7 | 79.4 |
> | 0.50 : 0.50 : 0.00 | 71.9 | - | - |
>
> We apologize for the insufficient explanation in the initial submission. The revised manuscript will include these ablation studies and a detailed discussion on the rationale behind the sampling ratio and its cross-dataset applicability, directly addressing your call for a deeper analysis.

---

> ### Author Response · Authors · 2025-11-20
> **Author Response for Reviewer 1MpC (3/3)**
>
> > ***Question 3:*** The paper's presentation could be improved for clarity. The three equations provided (Eq. 1-3) appear non-essential to the core idea and could be moved to the appendix. More importantly, the explanation of the core Refreshment and Retrieval mechanisms would benefit from a more intuitive, high-level explanation.
>
> We thank the reviewer for this valuable suggestion to improve clarity. We fully agree and have implemented the following revisions. As suggested, the equations (originally Eq. 1–3) have been moved to the appendix. More importantly, we have restructured the explanation of the core Refreshment and Retrieval mechanisms. They are now introduced with a more intuitive, high-level description before technical details are provided. Specifically, the Refreshment mechanism is introduced as selectively integrating high-confidence predictions into ego-pose-associated tiles of the global map, while the Retrieval mechanism is described as querying relevant map elements from specific tiles anchored to the ego pose. To further enhance clarity, we have also included pseudo-code for both mechanisms (see https://imgur.com/a/cUoOhzr) to illustrate their step-by-step workflows. We believe these changes significantly improve the readability and intuitive understanding of our approach, and we are grateful for the constructive feedback.
>
> > ***Question 4:*** The benefits of the temporal-prior are difficult to fully assess from static image results alone. The paper would be much stronger if it included supplementary videos to visually demonstrate the resulting temporal consistency and performance advantages.
>
> This is an excellent point, and we thank the reviewer for the suggestion. We have created supplementary videos to visually demonstrate the temporal consistency and performance advantages. The videos (see https://imgur.com/a/NCfhHwP; the first 72 of 150 full videos are included in the supplementary materials due to size constraints) provide a top-down comparison of Ground Truth, UniPrevPredMap (No Prior), UniPrevPredMap (Temporal Prior), and UniPrevPredMap (Temporal and Map Priors). The visual comparison clearly shows the enhanced temporal smoothness achieved with the temporal prior. Additionally, we will include final frames from representative sequences in the appendix to further illustrate the improvements. We appreciate this suggestion, which we believe strengthens the paper significantly.
>
> Once again, we are truly grateful for your time and insightful comments, which have been instrumental in guiding us toward a substantially improved manuscript. We hope our detailed responses and commitment to these revisions adequately alleviate your concerns.

---

### Official Review · Reviewer_qRog · 2025-11-01

**Soundness:** 2
**Presentation:** 2
**Contribution:** 2
**Rating:** 4
**Confidence:** 3

**Summary:**

This paper presents a unified prior-informed framework for online vectorized HD map construction, Uni-PrevPredMap. This framework integrates two complementary prior sources — previous predictions and corrupted HD maps — to enhance temporal consistency. Uni-PrevPredMap operates under a tri-mode paradigm, which ensures consistent behavior across non-prior, temporal-prior, and temporal-map-fusion modes. This paper also proposes the tile-indexed 3D vectorized global map processor, to manage prior information.
The framework is tested with nuScenes and Argover2 datasets showing some improvements over SOTA methods.

**Strengths:**

* The paper tackles an important challenge in online HD map construction by integrating temporal and prior information in a unified, adaptable framework.
* The proposed tri-mode paradigm is conceptually interesting, as it provides operational flexibility between prior-free and prior-informed scenarios.
* The empirical results suggest robustness to corrupted priors, demonstrating some progress in this specific research field.

**Weaknesses:**

* (L071) It would be helpful to briefly describe the “idealized fidelity assumptions” of maps early in the paper. As written, readers must infer these assumptions from context. A short explanation would improve clarity and motivation.

* The distinctions between non-prior, temporal-prior, and temporal-map-fusion modes remain somewhat unclear.
It would be useful to specify how each mode is derived or activated—for example, what features or signals determine which mode the system operates in.

* The tile-indexed 3D vectorized global map processor appears conceptually similar to the memory buffer mechanism in MapUnveiler (Kim et al., 2025) and MapTracker (Chen et al., 2024). A clearer explanation of the main differences and any improvements introduced by Uni-PrevPredMap would help readers better understand the novelty.

* The term “UTM coordinate” should be defined when first introduced. Not all readers may be familiar with it, and clarity here would avoid confusion.

* Although the experimental results generally show improvements, the model still underperforms compared to some SOTA methods (Tab.1). This should not diminish the paper’s overall contribution, but it would strengthen the work to discuss potential reasons for these results—e.g., trade-offs between robustness and accuracy, or the effects of noisy priors.

**Questions:**

Please refer to Weakness section.
Although this paper starts with an interesting approach, I was not able to fully understand the key concept, tri-mode paradigm, due to lack of detailed explanations. Also, current version requires further clarifications of its contributions with prior arts.

---

> ### Author Response · Authors · 2025-11-21
> **Author Response for Reviewer qRog (1/2)**
>
> Thank you for the constructive feedback and for recognizing the importance and conceptual interest of our work. We have carefully revised the manuscript to address all the concerns regarding conceptual clarity, definitions, comparative novelty, and result interpretation, as detailed below.
>
> > ***Question 1:*** It would be helpful to briefly describe the “idealized fidelity assumptions” of maps early in the paper. As written, readers must infer these assumptions from context. A short explanation would improve clarity and motivation.
>
> We appreciate this valuable suggestion. To improve motivation and readability, we have added a brief explanation early in the paper regarding the "idealized fidelity assumptions." Specifically, we now clarify that in this work, "idealized map fidelity" assumes the prior map is (a) geometrically accurate (with negligible errors), (b) semantically complete (containing all relevant map elements), and (c) fully covering the operational area. Our approach is designed to handle deviations from these assumptions, such as geometric noise, missing or extraneous map elements, and uncovered regions. This addition helps better frame the problem and underscores the contribution of our method.
>
> > ***Question 2:*** The distinctions between non-prior, temporal-prior, and temporal-map-fusion modes remain somewhat unclear. It would be useful to specify how each mode is derived or activated—for example, what features or signals determine which mode the system operates in.
>
> We thank the reviewer for highlighting this lack of clarity. In the revised manuscript, we have provided a more detailed explanation of the mode selection mechanism. Specifically:
>
> 1. During training, the three modes are randomly sampled according to a fixed ratio of 0.5:0.3:0.2. In the temporal-map-fusion mode, map vectors from both the temporal buffer and the corrupted map are retrieved from the global map, rasterized into temporal and map heatmaps, and then fed into the BEV encoder and query generator. In the temporal-prior mode, only map vectors from the temporal buffer are used to produce the temporal heatmap, while the map heatmap is initialized as a zero matrix. In the non-prior mode, both the temporal and map heatmaps are initialized as zero matrices.
>
> 2. During inference, the system adaptively selects the appropriate mode based on real-time data availability. Temporal-map-fusion mode is activated when operating continuously within a map-covered area where both the temporal buffer and corrupted map data are available. Temporal-prior mode is activated upon entering a region lacking map coverage, utilizing only the temporal buffer. Non-prior mode is activated when both the temporal buffer is unavailable (e.g., after a system restart) and map coverage is absent.
>
> We have clarified these activation criteria in the methodology section and will include a pseudo-code example (see https://imgur.com/a/EFvrSfc) in the appendix to illustrate the data flow for each mode unambiguously.

---

> ### Author Response · Authors · 2025-11-21
> **Author Response for Reviewer qRog (2/2)**
>
> > ***Question 3:*** The tile-indexed 3D vectorized global map processor appears conceptually similar to the memory buffer mechanism in MapUnveiler (Kim et al., 2025) and MapTracker (Chen et al., 2024). A clearer explanation of the main differences and any improvements introduced by Uni-PrevPredMap would help readers better understand the novelty.
>
> We appreciate the reviewer's question regarding the relationship between our method and related works like MapUnveiler and MapTracker. This provides a valuable opportunity to clarify the key distinctions. While all three approaches utilize a memory mechanism for map representation, there is a fundamental difference in design. Concisely, our memory buffer is prediction-based and region-centric, whereas those in MapUnveiler and MapTracker are feature-based and time-centric. Specifically: (a) MapUnveiler uses clip tokens to aggregate temporal information from historical and current BEV features, meaning its memory is built upon BEV features and organized in a time-centric manner. (b) MapTracker leverages historical BEV features and query features in a streaming fashion, resulting in a memory buffer that is also feature-based and time-centric. In contrast, our approach retrieves historical predictions and corrupted maps from a tile-indexed 3D vectorized global map, and then performs conditioned prediction. This makes our memory buffer prediction-based and region-centric. The advantages of our design are two-fold: (a) The prediction-based memory allows temporal buffers and corrupted maps to share a similar vectorized representation, enabling seamless integration of corrupted maps and supporting a unified framework. (b) The region-centric organization provides temporal priors not only from recent observations but also from predictions made by other vehicles nearby or from passes days ago, leveraging the static nature of map elements. We will revise the related work section to include a clearer technical comparison with MapUnveiler and MapTracker, and will emphasize these core contributions of Uni-PrevPredMap.
>
> > ***Question 4:*** The term “UTM coordinate” should be defined when first introduced. Not all readers may be familiar with it, and clarity here would avoid confusion.
>
> Thank you for this suggestion. We agree that defining the term improves readability. The manuscript has been revised to include a definition upon the first mention of "UTM coordinate", clarifying that it refers to the Universal Transverse Mercator system, a global coordinate system for precise location referencing.
>
> > ***Question 5:*** Although the experimental results generally show improvements, the model still underperforms compared to some SOTA methods (Tab.1). This should not diminish the paper’s overall contribution, but it would strengthen the work to discuss potential reasons for these results—e.g., trade-offs between robustness and accuracy, or the effects of noisy priors.
>
> Thank you for this constructive feedback. We acknowledge that while Uni-PrevPredMap shows strong overall performance, it does not outperform every SOTA method across all metrics, and we appreciate the opportunity to discuss the possible reasons.
>
> As shown in Table 1, in the map-absent setting, Uni-PrevPredMap achieves a Boundary AP that is 0.7 lower than Mask2Map. We attribute this primarily to the difference in conditioning strategies: Mask2Map uses the segmentation heatmap of the current frame as prior, which offers highly precise and up-to-date spatial information. This is particularly advantageous for modeling boundaries, which tend to exhibit higher inter-frame variation. In contrast, our approach relies on historical heatmaps, leading to a slight decrease in Boundary AP under map-absent conditions. That said, Mask2Map’s design involves higher computational cost, yielding an inference speed of only 9.5 FPS. More importantly, when a corrupted map prior is provided, Uni-PrevPredMap surpasses Mask2Map by a significant margin of 5.0 AP.
>
> In the case of pedestrian crossings, Uni-PrevPredMap underperforms MapTracker by 2.1 AP under the map-absent setting. This can be explained by the fact that pedestrian crossings are less structured and more challenging to reconstruct from rasterized heatmaps compared to dividers and boundaries. MapTracker leverages historical query features that preserve vector-level semantics, which helps reduce modeling difficulty for such elements. However, when a noisy map prior is introduced, the performance gap narrows to just 0.3 AP. Under clean map conditions, Uni-PrevPredMap outperforms MapTracker by 1.8 AP.
>
> We will incorporate the above discussion into the revised manuscript to better contextualize these results.
>
> We sincerely thank the reviewer for the insightful comments and the opportunity to improve our manuscript. The feedback has been invaluable in helping us enhance the clarity, rigor, and overall presentation of our work.

---

> > ### Comment · Reviewer_qRog · 2025-11-27
> >
> > I believe the authors have addressed most of my concerns and willing to increase the rating from 4 to 6.
> > From the rebuttal, I become more curious about whether the introduction of the the segmentation heatmap of the current frame as prior to the proposed approach would further improve the prediction performance as Mask2Map proposed.
> > I believe this may increase the computational resources, but would increase the reliability of the proposed method even under refined prior conditions. I'd like to hear the authors thoughts on this matter.

---

> > > ### Author Response · Authors · 2025-11-28
> > >
> > > We sincerely thank the reviewer for their supportive feedback and for raising the score. We are also grateful for the constructive suggestion regarding the use of a segmentation heatmap of the current frame as an additional prior.
> > >
> > > We agree that introducing a segmentation head after the BEV encoder and incorporating the resulting heatmap, together with temporal and map priors, to generate conditioned queries represents a natural and promising extension. Such an approach could enrich the model's prior representations. In particular, the segmentation heatmap offers forward-looking cues that may complement the temporal buffer and serve as a corrective reference when map data is incomplete or noisy, potentially improving both training stability and prediction accuracy.
> > >
> > > As the reviewer rightly noted, this enhancement would likely increase computational complexity. A thorough and fair evaluation would require integrating a high-quality segmentation network capable of real-time performance within our pipeline.
> > >
> > > We truly appreciate the reviewer's insightful suggestion, which reinforces and extends our own thinking in this valuable research direction.

---

### Author Response · Authors · 2025-11-26
**General Response to Area Chairs and Reviewers**

Dear Area Chairs and Reviewers,

We sincerely thank all the reviewers for their insightful and constructive feedback. We are encouraged by the positive recognition of our work, such as the reasonable design (uNTF), strong academic innovation (fBUh), interesting approach (1MpC), and addressing an important challenge (qRog) in the complementary unification of previous predictions and corrupted HD maps; the practical value (uNTF), strong applicability (fBUh), and conceptual interest (qRog) of the tri-mode paradigm; as well as the comprehensive experimental designs (fBUh) and empirically strong performance (1MpC, qRog).

In response to the valuable comments, we have thoroughly revised the manuscript. Key modifications, which are highlighted in yellow in the updated manuscript, address the core suggestions across the following areas:
1. **Improved Motivation and Presentation**: We have strengthened the abstract and introduction to more clearly position our work as the first to complementarily integrate previous predictions and corrupted HD maps. The notion of "idealized map assumptions" is now explicitly defined early in the paper to enhance motivational clarity, and the term "UTM" is defined upon its first use. The related work section has been refined to better distinguish our approach from existing methods, and the role of the tri-mode paradigm is more clearly elaborated in the ablation study.
2. **Improved Methodological Clarity**: We have provided a more accessible explanation of the tri-mode paradigm’s implementation, including a detailed description and accompanying pseudo-code (provided in the appendix). An overview now precedes the refreshment and retrieval mechanisms to offer better conceptual grounding, while non-essential equations have been moved to the appendix along with corresponding algorithmic pseudo-code.
3. **Expanded Experimental Analysis**: The results and discussion have been deepened with additional analysis cases, particularly where performance lags behind certain state-of-the-art methods (e.g., in boundary and pedestrian crossing AP), to objectively acknowledge trade-offs and inspire future research. Further ablation studies on the tri-mode sampling ratio have been added to demonstrate its empirical basis and cross-dataset consistency.
4. **Refined Discussion of Limitations and Future Work**: We have clarified the two practical reasons limiting the use of 3D priors in the current design, outlining a clearer path for future improvements. We also explicitly acknowledge the limitation of not using real-world corrupted maps (e.g., outdated or crowdsourced HD maps) and emphasize the need to establish standardized benchmarks for realistic map corruptions.
5. **More Comprehensive Results Presentation**: We have expanded the appendix to include three new elements: a computation cost analysis of the 3D tile-indexed vectorized global map processor for a more thorough evaluation, a speed-accuracy trade-off comparison to better contextualize our performance, and video visualizations to more clearly demonstrate the advantages of temporal priors.

We believe these revisions have significantly strengthened the paper's clarity, rigor, and overall contribution. We thank the reviewers again for their time and effort, which have been invaluable in improving this work.

With heartfelt gratitude and warmest regards,

The Authors

---

### Meta-Review · Area_Chair_xWUQ · 2025-12-21

**Summary:**

This work showcases a vectorized HD map construction framework that ensures the maintenance of operational consistency. It integrates temporal prior information (essentially, previous predictions) and a corrupted map prior to improve robustness of the constructed maps. It is evaluated on the nuScenes and Argoverse2 benchmarks and achieves state-of-the-art performance in its evaluation setting.

**Reviewer Concerns:**

The main reviewer concerns are focused on clarity (qRog, 1MpC, fBUh), performance (qRog, uNTF), and some conceptual limitations (fBUh, uNTF).

**Reviewer Scores:**

Overall, the authors responses addressed many of the reviewers concerns on clarity although some may require another round of review and more detailed changes. The conceptual limitations of the work were only addressed in so far as that the authors admitted the limitations and committed to mentioning them. The verdict on the perpformance improvement is mixed. One reviewer did not expect the authors to outpreform SOTA even further. For another reviewer it seems not clear whether the response would have been sufficient. Ultimately, if the performance improvement is small because the benchmark is saturated, it makes you wonder about either the usefulness of the method or the suitability of the benchmark.

---

### Decision · Program_Chairs · 2026-01-26

Reject